# RF-Agent: Automated Reward Function Design via Language Agent Tree Search

**Ning Gao, Xiuhui Zhang, Xingyu Jiang,**
**Mukang You, Mohan Zhang, Yue Deng***

Beihang University
37 Xueyuan Road, Haidian District, Beijing
{gaoning_ai, zhangxiuhui, jxy33zrhd}@buaa.edu.cn
youmukang@gmail.com, {zmh666, ydeng}@buaa.edu.cn

## Abstract

Designing efficient reward functions for low-level control tasks is a challenging problem. Recent research aims to reduce reliance on expert experience by using Large Language Models (LLMs) with task information to generate dense reward functions. These methods typically rely on training results as feedback, iteratively generating new reward functions with greedy or evolutionary algorithms. However, they suffer from poor utilization of historical feedback and inefficient search, resulting in limited improvements in complex control tasks. To address this challenge, we propose RF-Agent, a framework that treats LLMs as language agents and frames reward function design as a sequential decision-making process, enhancing optimization through better contextual reasoning. RF-Agent integrates Monte Carlo Tree Search (MCTS) to manage the reward design and optimization process, leveraging the multi-stage contextual reasoning ability of LLMs. This approach better utilizes historical information and improves search efficiency to identify promising reward functions. Outstanding experimental results in 17 diverse low-level control tasks demonstrate the effectiveness of our method. The source code is available at https://github.com/deng-ai-lab/RF-Agent.

## 1 Introduction

Reward design is a crucial component in Reinforcement Learning (RL), significantly affecting the performance and training efficiency of the policy, especially in low-level control tasks like locomotion[40] and complex manipulation[3, 15]. Although task evaluation metrics (*e.g.*, success rate, movement speed) can be directly used as rewards, their sparse or one-dimensional nature presents challenges for policy optimization. Therefore, reward shaping[34] through dense reward functions is crucial for guiding policy learning more efficiently[46]. The most common approach involves human experts manually crafting a dense reward function, which is interpretable but requires extensive expertise and may be suboptimal[4]. To address this, Inverse RL[65, 54, 13] and Preference-based RL[10, 22, 35] attempt to generate dense rewards by learning reward models from expert demonstrations or preferences. However, they still rely on large amounts of expert data and lack interpretability.

Recently, large language models (LLMs) have excelled in not only traditional NLP tasks[33] but also logical reasoning and agent-based tasks[50, 58]. Given their impressive world knowledge [52] and coding capabilities [56, 60], a natural thought is to leverage LLMs to generate interpretable dense reward functions, replacing expert effort. Approaches like L2R[59] and Text2Reward[55]

---

*Corresponding author

39th Conference on Neural Information Processing Systems (NeurIPS 2025).

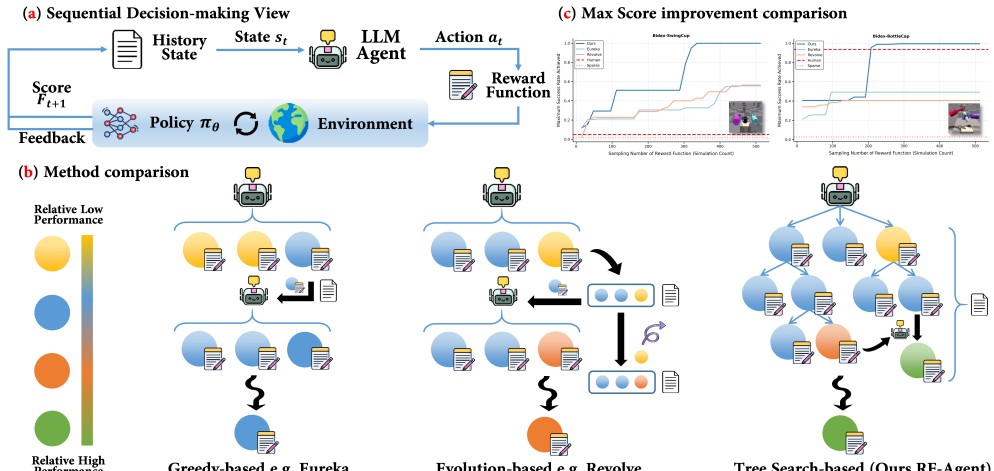

Figure 1: Quick view. (a) Reshaping LLM-based Reward Function Design Problem from sequential decision-making view. (b) Abstract pipeline comparison between different approaches. (c) Comparison between different methods on max success rates with the numbers of complete training times.

made strides in this direction, achieving remarkable results. In addition, more recent methods[28, 17, 61, 62, 18] have further advanced the field by combining more refined feedback design and evolutionary concepts. We view these methods as part of a sequential decision process, as shown in Fig1.a. The interaction between the policy and environment forms an internal system, where the LLM, as a language agent, continuously interacts with this system. The action of the LLM represents the process of generating a reward function, and the internal system provides results after training as feedback based on the action. The action and feedback in the current decision-making process are finally integrated into the history state and utilized by LLM for future actions.

From the sequential decision-making perspective, current research can be categorized into two reward generation modes, as shown in Fig1.b. The first mode, exemplified by Eureka[28], follows a greedy algorithm[48] approach, where the best reward function is retained after a batch generation of reward functions and serves as the starting point for the next iteration. The second mode, represented by Revolve[17], adopts an evolutionary algorithm[11] approach, maintaining a population and evolving new reward functions by integrating individual information from the population. However, when utilizing LLMs to design reward functions for complex control tasks[9], the performance improvement of these methods remains limited, as illustrated in Fig1.c. This limitation hinders the generation of high-quality and training-efficient reward functions through LLMs, constraining the performance of this pipeline. We identify two key factors leading to limited performance improvement. First, from the search perspective, greedy or evolutionary algorithms struggle in complex decision-making problems, as they fail to effectively balance exploitation and exploration, often resulting in premature convergence to local optima or excessive exploration of possibilities[37]. Second, regarding information usage, existing methods only retain local historical information, overlooking the potential decision paths that could transition from low-performing to high-performing reward functions (Fig1.b ours).

Based on the analysis above, we introduce a novel reward function design framework, RF-Agent, which treats the LLM as an agent to address the challenges outlined by enhancing their reasoning and decision-making capabilities. Specifically, we employ a tree structure to represent the entire reward design process (Fig1.b ours) and its history states. Each node in this tree corresponds to a distinct decision strategy and its associated reward function, effectively leveraging the multi-stage contextual reasoning capabilities of LLMs. To further improve the framework, we incorporate the Monte Carlo Tree Search (MCTS) algorithm[5] to balance exploration and exploitation within the decision space. Additionally, during the action expansion phase, we integrate a set of heuristic action types to guide the LLM in generating reward functions with varying inclinations. This enables RF-Agent to appropriately incorporate global decision information from other paths, ensuring the full utilization and refinement of promising reward functions. Finally, environmental feedback and self-verify metrics contribute to an overall evaluation, which forms a value prior for node selection, thereby enhancing the search optimization capabilities of the entire RF-Agent framework.

We evaluate the proposed RF-Agent across 17 tasks in IsaacGym[31] and Bi-DexHands[9], covering task types such as locomotion and robot arm/hand manipulation. The results indicate that our approach outperforms the current state-of-the-art LLM-based reward function design methods, producing high-performance and training-efficient reward functions. Such achievement demonstrates the potential of leveraging language agents in combination with reasoning and search frameworks for reward function design. Ablation studies further validate the effectiveness of our architecture.

## 2  Related Works

**Reward Design with LLMs.** Reward engineering has long been a challenge in reinforcement learning[45, 26], with the common approach being manual creation of reward functions by experts[23, 4]. Inverse RL[1, 19, 65, 54, 13] and Preference-based RL[10, 22, 35] generate rewards through learned models, but still require expert demonstrations and lack interpretability. Recent advancements leverage LLMs, which exhibit strong world knowledge and logical reasoning[58, 14], in tasks like code generation[8, 39] and robotic control planning[27, 44, 49]. Some works have extended this to generating rewards for RL. For instance, [24] uses LLMs to generate binary rewards based on user intent in negotiation games, and [12] calculates rewards based on the similarity between state transitions and goals generated by LLMs. However, these methods suffer from excessive LLM calls and inconsistencies. In contrast, L2R[59] and T2R[55] generate dense, interpretable reward functions, while T2R directly offers plug-and-play reward function Python code, introducing a promising paradigm for reward engineering. Recent works like Eureka[28] use greedy iterative approaches to optimize reward functions, while Revolve[17] incorporates evolutionary algorithms to evolve new rewards. However, both methods still face challenges with inefficient use of historical feedback and low search efficiency. Our RF-Agent addresses these issues by introducing Monte Carlo Tree Search to manage reward function design and optimization process from a decision-making perspective. It fully utilizes LLMs' contextual reasoning and historical feedback, thereby significantly enhancing the performance of the designed reward function.

**LLM-based Agent for Decision Making.** The world knowledge and logical reasoning abilities of LLMs enable them to serve as agents for solving decision-making problems[36, 49, 6]. For example, LLMs have been used as high-level controllers in robotic planning [2], interacted with web browsers to fulfill user needs [32], and tackled multi-step decisions in text-based adventure games [43]. As tasks become more complex, methods such as Twosome[47] attempt to fine-tune LLM-based agents to interact directly with the environment, but this often incurs high training costs. In contrast, training-free approaches that utilize multi-step reasoning to leverage LLMs' contextual learning have gained popularity[53, 51, 58, 63, 38]. This "Test-time Compute" paradigm [25] enhances reasoning to influence decision performance, with Chain-of-Thought (CoT)[53] being the most prominent method. ReAct[58] incorporates CoT into decision-making, while self-refine[29] and Reflexion[42] add self-feedback mechanisms. However, these linear methods neglect alternative decision paths. ToT[57] breaks the decision process down using tree search, and methods like LATS[64] and RAP[16] further introduce Monte Carlo Tree Search (MCTS) to enhance reasoning during decision-making. Inspired by these approaches, we frame reward function design as a decision-making process, enhancing the logical reasoning of LLM to better analyze historical feedback and generate high-quality reward functions.

## 3  Problem Setting

In standard reinforcement learning (RL) [46] tasks, the problem is commonly modeled as a Markov Decision Process (MDP), formalized by the tuple $\mathcal{M} = \langle S, A, T, R, \gamma, \rho_0 \rangle$. In this formulation, $S$ denotes the state space and $A$ represents the action space. $\gamma$ is the discount factor, and $\rho_0$ is the initial state distribution. The environment's dynamics are described by the transition function $T : S \times A \times S \to [0, 1]$, which specifies the probability of transitioning to a new state given the current state and action. The reward function $R : S \times A \to \mathbb{R}$ defines the immediate reward received after taking an action in a given state.

We then introduce the **Reward Design Problem** (RDP)[45], given by the tuple $P = \{M, \mathcal{R}, \pi, F\}$, where $M = (S, A, T)$ is the environment world model including state space $S$, action space $A$, and transition function $T$. $\mathcal{R}$ indicates the space of all reward functions where each $R \in \mathcal{R}$ is a reward function that maps a state-action pair to a scalar $R : S \times A \to \mathbb{R}$. $\pi : S \to A$ indicates a

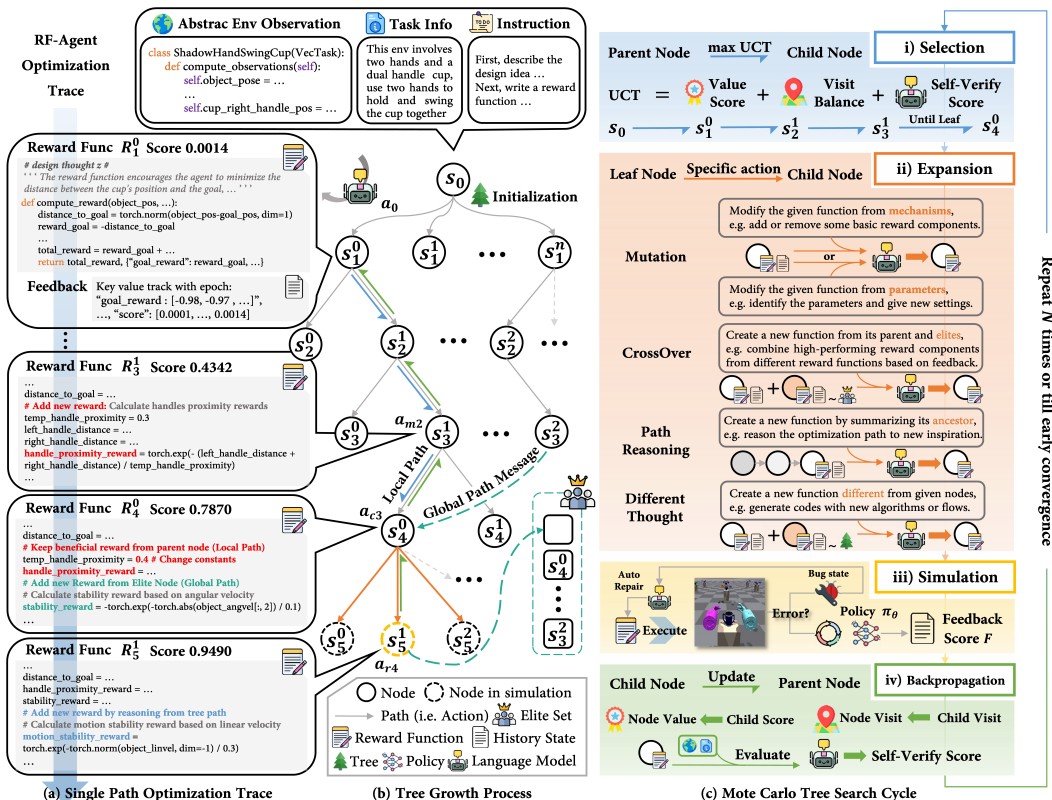

Figure 2: Illustration of our RF-Agent with (a) an exemplary reward function optimization path from the tree, (b) total tree growth process, and (c) iteration based on MCTS. Please refer to the Appendix B for prompts used in RF-Agent, mainly including initialization, expansion, and other process.

policy sampled form the policy space $\Pi$ and $F(\cdot) : \Pi \to \mathbb{R}$ is the evaluation metric that measures the real performance of $\pi$ on a given task. Given a reward function $R$, the tuple $(M, R)$ forms a Markov Decision Process $\mathcal{M}$. We additionally denote $\mathscr{A}_M(\cdot) : \mathcal{R} \to \Pi$ as a learning algorithm (*e.g.*, PPO[41]) that outputs the policy $\pi$ under the given MDP $(M, R)$. The goal of the RDP is to output a reward function $R \in \mathcal{R}$ such that the policy $\pi := \mathscr{A}_M(R)$ under the MDP $(M, R)$ achieves the highest evaluation score $F(\pi)$.

In our **Reward Function Design Problem** (RFDP) setting, each $R \in \mathcal{R}$ is specified as Python code and given by a reward designer $G : \mathcal{L} \to \mathcal{R}$ according to the input string $L$ as instruction. Here we choose a pre-trained language model $p_\theta(\cdot)$ as the reward function designer $G$. After the policy $\pi$ completes the training of the fix number of steps under the given reward function $R$ and learning algorithm $\mathscr{A}_M$, we will receive feedback from the environment about this learning process. In addition to the evaluation score $F$, a string $l_{feedback}$ as language feedback will indicate the execution situation. Finally, given a string $l_{task}$ that specified the task, the objective of the RFDP is to output a reward function code $R \sim p_\theta(l)$ such that $F(\mathscr{A}_M(R))$ is maximized.

# 4 Method

In this section, we first explore the challenges associated with solving the LLM-based RFDP problem, with a particular emphasis on the in-context learning capabilities of LLMs. We then introduce RF-Agent, a framework that enhances the multi-stage contextual reasoning of LLMs by integrating Monte Carlo Tree Search (MCTS), thereby improving the efficiency of reward function search through better utilization of historical feedback and evaluation metrics. Finally, we show an exemplary reward function optimization process to illustrate the effectiveness of RF-Agent.

## 4.1 Rethinking LLM-Based Reward Function Design Problem

From a decision-making perspective, each reward function $R$ generated by an LLM $p_\theta(l)$ can be viewed as an action $a$, with the corresponding input instruction $l$ as the state $s$. This presents a key challenge in RFDP: the state and action space defined on language is vast, while the evaluation metrics $F$ and feedback $l_{feedback}$ provided by the environment after training are sparse and text-based. As a result, using feedback to directly optimize $p_\theta(\cdot)$ via gradients is impractical and costly.

The in-context learning ability of LLMs offers a promising alternative to solve the above problem. In many text-based tasks, LLMs act as language agents, leveraging varying state information as context to iteratively make decisions[58, 64]. Based on the analysis, we identify that previous LLM-based RFDP methods simply treat LLM contextual learning as a basic usage, failing to integrate it effectively within the RFDP framework. These methods suffer from limited historical information utilization and low search efficiency to generate high-performance reward functions. To address these issues, we propose the RF-Agent framework, which fully exploits multi-stage contextual reasoning of LLM to better utilize evaluation metrics and textual feedback in RFDP and improve the efficiency of reward function search.

## 4.2 RF-Agent for Reward Function Design

**Overview.** Our RF-Agent is illustrated in Fig2. We structure the optimization to solve the LLM-based RFDP as a Monte Carlo Tree Search process, with the root node $n_{root}$ serving as a virtual node. The remaining nodes each represent a reward function $R$ with its training feedback. Based on classical MCTS, we first generate an initial set of child nodes, after which the RFDP solution process consists of four stages: selection, expansion, simulation, and backpropagation. These stages iterate until the maximum iteration limit $N$ is reached or early convergence. In the selection phase, we select the most promising node based on an improved Upper Confidence Bound for Trees (UCT), which incorporates evaluation metrics, self-verify metrics, and exploration count. During the expansion phase, RF-Agent utilizes specific actions to guide the LLM in generating reward functions for child nodes, ensuring effective exploration of the reward function space. Finally, in the simulation and backpropagation phase, RF-Agent trains policies with generated reward functions and receives feedback to update the node states for next selection. See Appendix G for the algorithm of RF-Agent.

**Initialization.** During the initialization phase, in addition to the brief task information $l_{task}$, RF-Agent is only granted access to the observational part of the environment code, $l_{obs}$, consistent with previous approaches[28, 17]. We then provide both inputs to the LLM, leveraging its world knowledge and zero-shot instruction-following capabilities to generate the reward function:

$$R \sim p_\theta(x, z) \,, \; z \sim p_\theta(x) \,, \; \text{where } x = [l_{task}, l_{obs}, l_{action}] \tag{1}$$

Here, $l_{action}$ refers to the instructions, and $z$ represents the design thought in the reasoning process. As shown in Fig.2.a, we prompt the LLM to first generate the design thought and then create the specific reward function based on it, leveraging the advantages of logical reasoning[53].

The generated reward functions are then integrated with the environment to train the policy under the learning algorithm $\mathscr{A}_M$. The process returns an evaluation metric $F$ (*e.g.*, task success rate) and feedback $l_{feedback}$, which tracks the changes of key components in $R$ during training as shown in App. B.2. Then, the construction from the root node $n_{root}$ to the first layer of child nodes is complete, with each node state represented as $s = [z, R, F, l_{feedback}]$. The tree structure stores all of the node states throughout the development, enabling their full utilization in the selection and expansion stages.

**Selection to the Potential Node.** In the selection stage, RF-Agent balances exploration and exploitation to identify promising nodes as shown in Fig2.c.i and blue arrow in b, improving search efficiency for promising reward functions. We complete this stage by sequentially selecting child nodes with the highest UCT values until a leaf node. In the RFDP setting, each node maintains a value $Q(s)$ reflecting the score of its reward function and those of its descendants, along with a visit count $N(s)$.

Additionally, RF-Agent introduces a self-verify score to improve selection. In complex control tasks, early-generated reward functions often have low scores $F$, even nearly zero, resulting in the value $Q$ constructed based on $F$ not being able to effectively reflect the potential of the node. Inspired by the inherent evaluation capabilities of LLMs[42], we prompt the LLM to first analyze how an expert completes the task, then output a score representing the likelihood of obtaining an expert-level policy

by training under the current $R$ as shown in App. B.4. The final UCT calculation is as follows:

$$\text{UCT}(s_{child}) = \frac{Q(s_{child}) - Q_{min}}{Q_{max} - Q_{min}} + \lambda \cdot (\sqrt{\frac{2 \cdot \log(N(s_{parent}) + 1)}{N(s_{child})}} + \sigma(v_{self}(s_{child}))) \quad (2)$$

Here, we normalize the $Q$-value to enhance the robustness of the UCT calculation in different tasks. The parameter $\lambda$ controls the weight of the exploration term and linearly decays during the search to promote early exploration while ensuring later convergence. $v_{self}$ represents the self-verify score scaling with softmax $\sigma$ and also decays with $\lambda$.

**Expansion with LLM-based actions.** In the expansion stage, RF-Agent incorporates historical information to guide the LLM in generating new reward functions similar to Eq.1, but with different $l_{action}$. Previous methods[28] typically regenerate reward functions based on a single prompt and limited historical data, thus cannot fully leverage the LLM in-context learning abilities and limit effective exploration of the reward function space. When humans tackle problems, a complete decision-making process usually involves various types of action, with different actions employed under different states. Inspired by this, RF-Agent enhances exploration of the reward function space by defining different action types to generate the reward functions with utilization of historical information from the whole tree, as shown in Fig2.c.ii and orange arrow in b. See App. B.3 and I.1 for detailed prompts and effects. The action types including $[a_{m1}, a_{m2}, a_{c3}, a_{r4}, a_{d5}]$ are as follows:

◇ *Mutation $a_{m1}$ & $a_{m2}$ : $l_{action} = l_{mutation}(s_{parent})$.* We introduce the mutation to represent local modifications to the existing reward function. Given a parent node, the mutation action guides the LLM to analyze the state of the parent node to optimize its reward function. Specifically, $a_{m1}$ guides the LLM to modify the structure of the reward function, such as adding or removing a component, while $a_{m2}$ focuses on adjusting the parameter weights within the reward function.

◇ *Crossover $a_{c3}$ : $l_{action} = l_{crossover}(s_{parent}, s_{node\sim\{elites\}})$.* To enhance the utilization of global historical information and accelerate the search process, we introduce a crossover operation to extract information from high-performance nodes. RF-Agent maintains an elite set, dynamically storing high-performance nodes from the tree based on evaluation scores $F$. We sample $k$ nodes from the elite set, weighted by their scores, and input them along with the parent node state to the LLM. Then RF-Agent guides the LLM to analyze the feedback and extract useful reward components from these nodes, combining them to form a new reward function.

◇ *Path Reasoning $a_{r4}$ : $l_{action} = l_{reason}(s_{parent}, s_{ancestor})$.* Each path from the root node to a leaf node represents a unique reward function optimization trace. Similar to how humans recall past experiences when solving new problems, RF-Agent leverages the knowledge in the reward function optimization trace by reasoning. By truncating a $k$-length tree path from the current node to the root, RF-Agent guides the LLM to reason the reward function optimization process, identifying the design strengths and further generating a new reward function.

◇ *Different Thought $a_{d5}$ : $l_{action} = l_{differ}(s_{parent}, s_{node\sim\{tree\}})$.* To prevent premature convergence during the search and enhance exploration of the reward function space, we introduce $a_{d5}$ to generate thoughts distinct from existing reward functions in the tree. RF-Agent randomly selects nodes from different paths, combines them with the parent node, and guide the LLM to generate a reward function structurally different from those in the selected nodes.

**Simulation with Reward Function.** In the simulation stage, we train the policy using the reward function R of expanded nodes, as shown in Fig. 2.c.iii. Generally, $R$ can be executed at once, then a complete node $s = [z, R, F, l_{feedback}]$ will be established. In complex environments, however, early LLM-generated reward functions may encounter execution errors. In such cases, we extract the traceback error and prompt the LLM to adjust $R$. This adjustment may cause the reward function $\hat{R}$ to diverge from the original design thought $z$, and even without adjustments, LLM hallucinations can misalign $R$ and $z$[20]. Since expansion actions are partly based on $z$, its accuracy is crucial. To address this, RF-Agent introduces a simple yet effective thought alignment process: during expansion, we generate $z$ before $R$; in simulation, once the reward function compiles correctly, we regenerate a more complete design idea $z_{new}$ based on $R$ or $\hat{R}$. See Appendix B.4 and I.2 for details.

**Backup with Evaluation.** The purpose of the backpropagation stage is to update the value $Q(s)$ and visit count $N(s)$, with generating a self-verify score $v_{self}$, as shown in Fig2.c.iv and green arrow in b. The process and motivation for generating the self-verify score are detailed in the expansion part,

while the update process for $Q(s)$ and $N(s)$ with update rate $\eta$ is as follows:

$$Q(s) = (1 - \eta) \cdot Q(s) + \eta \cdot \max_{s' \in \text{Child}(s)} Q(s') \; ; \; N(s) = \sum_{s' \in \text{Child}(s)} N(s') \tag{3}$$

### 4.3 Optimization of Reward Function in RF-Agent

As shown in Fig2.a, we illustrate an exemplary reward function optimization path for Swing Cup task. During early exploration, RF-Agent tends to expand extensively to try different reward functions such as using mutation actions to add the handle distance reward in $R_3^1$. Later, RF-Agent focuses on exploiting a few high-performance paths, such as continuing with $R_3^1$ by crossover of elite nodes to generate $R_4^0$, which modifies parameters and incorporates angular velocity stabilization rewards. Further optimization through path reasoning led to $R_5^1$, which adds a linear velocity stabilization reward and achieves high performance. See Appendix F.3 for more optimization examples.

## 5 Experiments

### 5.1 Experiment Environments

We test RF-Agent in two low-level control environments: IsaacGym[31] and Bi-DexHands[9], encompassing 8 control agents and 17 diverse tasks. We first evaluates on 7 representative tasks across locomotion and robot arm/hand manipulation from IsaacGym. To further test our method on more complex tasks, we selected 10 tasks from the Bi-DexHands, involving dual-arm manipulation with more intricate actions such as door closing, cup rotation, and bottle cap twisting. We divided these 10 tasks into two groups, expert-easy and expert-hard, based on human reward function success rates, enabling a comprehensive comparison of different LLM-based RFDP methods and human performance. Task details are provided in the Appendix A.

### 5.2 Baselines

**Human.** These use dense reward functions from the benchmark, written by reinforcement learning researchers who designed the tasks, representing expert-level reward engineering techniques.

**Sparse.** These are equivalent to using the evaluation score $F$, which directly reflects task completion quality, as the reward. This reward is typically sparse or single-dimensional.

**Eureka**[28] automatically generates dense reward functions by combining task information and environmental observations. More importantly, it generates rewards in batches and retains the best one based on feedback, optimizing the reward function through a greedy iterative approach.

**Revolve**[17] maintains a population and applies evolutionary operations to modify the reward functions with human-in-the-loop feedback. In our fully automated reward function design scenario, we implemented the Revolve auto version, where feedback is also provided by the environment.

### 5.3 Training Setup

**Policy Training.** Following Eureka, all tasks are implemented with a well-tuned PPO[30] and default hyperparameters provided by the benchmark. We tested the final reward functions through individual training with 5 different seeds and measured the quality of the reward functions by reporting the average maximum evaluation score achieved at each policy checkpoint.

**Evaluation metrics.** In the Bidex tasks, the success of a task is determined by a binary 0-1 signal, thus the evaluation score directly based on the success rate. In the Isaac tasks, the metric depends on its goal. For example, the goal of the Ant task is to move as quickly as possible, so the metric is set as the forward speed. See appendix for detailed task descriptions and evaluation scores.

**Method Implementation.** Since Eureka, Revolve, and RF-Agent optimize reward functions through sampling, we ensure a fair comparison by setting the same total sampling number of reward functions. In IsaacGym, we set the total limit to 80 and use the GPT-4o-mini-0718 and GPT-4o-0806 models for LLM implementation[21]. For the Bidex tasks, to evaluate the search performance of the method itself under complex control tasks, we increase the upper limit to 512 and use only the GPT-4o-mini

Table 1: Quantitative evaluations on IsaacGym. **Bold** indicates the best results within the same group. Avg norm score represents the average performance after calculating human normalized scores $(Method - Sparse)/(Human - Sparse)$ for each task.

| task | locomotion | | | | manipulation | | | Avg norm score |
|---|---|---|---|---|---|---|---|---|
| | Ant | Anymal | Humanoid | Quadcopter | AllegroHand | FrankaCabinet | ShadowHand | |
| Sparse | $0.14_{\pm0.05}$ | $-2.05_{\pm0.24}$ | $3.01_{\pm1.99}$ | $-1.35_{\pm0.04}$ | $0.03_{\pm0.00}$ | $0.04_{\pm0.08}$ | $0.04_{\pm0.00}$ | 0 |
| Human | $6.75_{\pm0.30}$ | $-0.03_{\pm0.00}$ | $5.89_{\pm0.81}$ | $-0.08_{\pm0.03}$ | $11.41_{\pm1.62}$ | $0.11_{\pm0.07}$ | $8.56_{\pm1.23}$ | 1 |
| LLM-based Reward Function Design with GPT-4o-mini | | | | | | | | |
| Eureka | $4.87_{\pm1.34}$ | $-0.67_{\pm0.04}$ | $3.29_{\pm0.19}$ | $-0.07_{\pm0.02}$ | $13.87_{\pm3.04}$ | $0.01_{\pm0.00}$ | $10.81_{\pm1.06}$ | 0.63 |
| Revolve | $5.16_{\pm1.37}$ | $-0.80_{\pm0.09}$ | $3.12_{\pm0.33}$ | $-0.08_{\pm0.02}$ | $18.40_{\pm0.97}$ | $0.19_{\pm0.17}$ | $10.61_{\pm1.38}$ | 0.67 |
| Ours | $\mathbf{6.22_{\pm0.90}}$ | $\mathbf{-0.01_{\pm0.00}}$ | $\mathbf{5.91_{\pm1.21}}$ | $\mathbf{-0.03_{\pm0.00}}$ | $\mathbf{22.52_{\pm1.83}}$ | $\mathbf{0.35_{\pm0.15}}$ | $\mathbf{13.13_{\pm0.82}}$ | **1.70** |
| LLM-based Reward Function Design with GPT-4o | | | | | | | | |
| Eureka | $5.77_{\pm0.85}$ | $-0.01_{\pm0.00}$ | $5.44_{\pm0.88}$ | $-0.03_{\pm0.02}$ | $22.55_{\pm0.99}$ | $0.52_{\pm0.21}$ | $12.27_{\pm0.91}$ | 2.00 |
| Revolve | $5.90_{\pm0.82}$ | $-0.01_{\pm0.00}$ | $4.87_{\pm0.23}$ | $-0.04_{\pm0.02}$ | $23.11_{\pm1.10}$ | $0.57_{\pm0.50}$ | $9.02_{\pm1.00}$ | 2.03 |
| Ours | $\mathbf{7.10_{\pm0.11}}$ | $\mathbf{-0.01_{\pm0.00}}$ | $\mathbf{6.37_{\pm0.32}}$ | $\mathbf{-0.03_{\pm0.01}}$ | $\mathbf{24.04_{\pm1.41}}$ | $\mathbf{0.79_{\pm0.18}}$ | $\mathbf{14.09_{\pm0.40}}$ | **2.68** |

model. For Eureka and Revolve deployment details, please refer to the Appendix D. As for the configuration of our RF-Agent, please refer to the Appendix C.

## 5.4 Experiment Results

**Advantages of RF-Agent in multi-class control tasks.** Tab.1 reports the absolute scores with standard deviation and average normalized performance of different methods on each IsaacGym task. We observe the following advantages:

(1) Our method outperforms other LLM-based reward function design methods across various tasks and control types, demonstrating RF-Agent's ability to leverage LLMs for reward function generation, with this advantage unaffected by changes in the LLM backbone model.

(2) Even with a lightweight LLM backbone model 4o-mini, RF-Agent-generated reward functions maintain superiority over human expert methods in nearly all tasks, while Eureka and Revolve do not, particularly in locomotion tasks.

(3) With a more powerful LLM backbone 4o, almost all methods achieve better reward functions within a limited number of training iterations. However, it is worth noting that Eureka experiences oscillations on FrankaCabinet, indicating its inability to generate effective reward functions with a lightweight model. Revolve fails to optimize performance on ShadowHand, while RF-Agent's performance improvement remains stable across all tasks.

These advantages highlight RF-Agent's effective use of contextual reasoning and historical information, enabling superior search efficiency for high-performance reward functions. Additionally, we further modify two control tasks to validate the generalization capabilities of different methods in out-of-distribution scenarios. Detailed results can be found in the AppendixF.4.

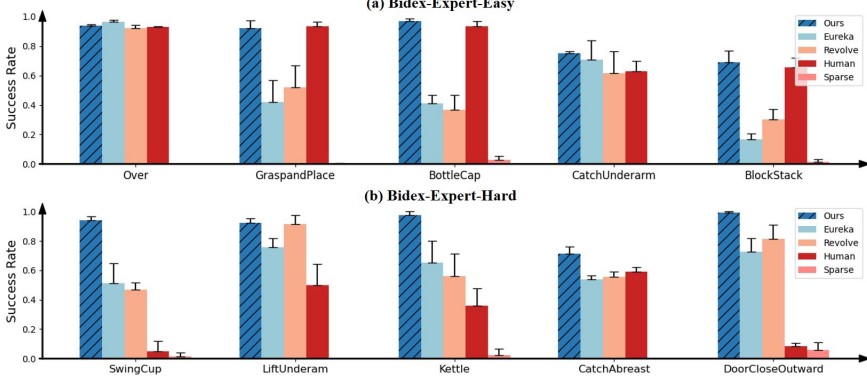

Figure 3: Success rates comparison with standard deviation upper bar on BiDex Expert-Easy/Hard.

**High Performance of RF-Agent under complex manipulation.** Fig.3 presents the results of different methods on BidexHands, with tasks divided into Expert-Easy and Expert-Hard groups based on human reward function outcomes for easier comparison. As shown, RF-Agent maintains a clear performance advantage over both human experts and other LLM-based reward function design methods, particularly on more complex tasks. Specifically, in the Expert-Easy tasks, where human reward functions have high success rates and smaller objects (*e.g.*, cubes, bottle caps) are manipulated, Eureka and Revolve fail to reach half of human performance on tasks like GraspAndPlace, BottleCap, and BlockStack. In contrast, RF-Agent matches or slightly exceeds human performance. In the Expert-Hard tasks, where human success rates are lower and larger objects (*e.g.*, kettles, doors) are manipulated, LLM-based methods perform all well, with RF-Agent showing an obvious advantage in most tasks. These results demonstrate that RF-Agent can design effective reward functions even in more complex task environments.

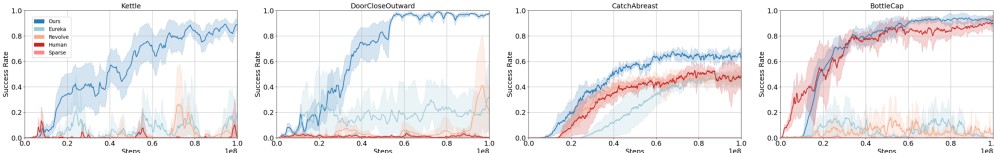

Figure 4: Success rates with exploration steps under reward functions by different methods.

**RF-Agent designs training-efficient reward functions.** Fig.4 shows the training curves for different reward functions on BiDexHands. Compared to Eureka and Revolve, Reward functions generated by RF-Agent can efficiently train the policy to converge to a higher success rate range, illustrating the high-quality reward function generation ability of RF-Agent. See Appendix F.1 for curves of others.

**The powerful search improvement capabilities of RF-Agent.** Fig.5 shows the average maximum scores achieved across tasks as reward functions are iteratively generated, reflecting the actual improvement of reward functions. In both Expert-Easy/Hard tasks, RF-Agent demonstrates high optimization efficiency, significantly outperforming other methods. These results indicate that RF-Agent effectively addresses the issue of limited reward function improvement in complex control. We also provide the maximum reward achieved on each single task in Appendix F.2.

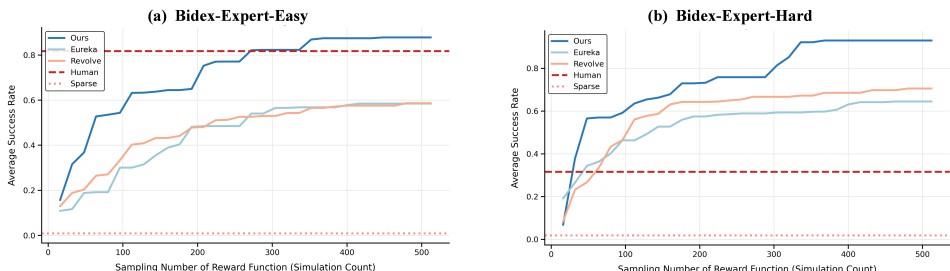

Figure 5: Reward function optimization performance with sampling counts.

## 5.5 Ablation Studies.

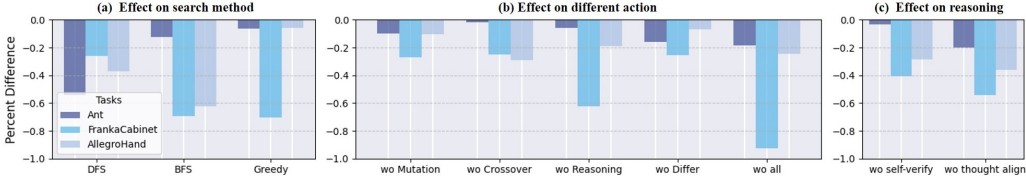

Figure 6: Ablations on the search method, action types, and reasoning component. We report the percent difference in performance compared to the complete RF-Agent. The experiment is built on Ant, FrankaCabinet, and AllegroHand to cover three task types with 4o-mini as the designer.

**Search Method.** To evaluate the impact of balancing exploration and exploitation on RF-Agent, we replace the improved UCT-based selection process with common search methods such as DFS, BFS,

and Greedy. As shown in Fig 6.a, each method experienced significant performance loss on at least one task, highlighting the importance of balance for search efficiency.

**Action types.** To evaluate the impact of the actions designed in the expansion stage, we relatively replaced the different action types with a basic action, providing only the parent node state to generate new reward functions. In Fig 6.b, each action type holds its value, and removing all actions leads to a significant degradation. This demonstrates that the action design of RF-Agent effectively leverages historical information and LLM in-context learning ability to generate high-quality reward functions. Additionally, we further investigate the impact of different action combinations on the experimental results. Detailed results and explanations can be found in the AppendixF.5.

**Reasoning.** To evaluate the impact of LLM reasoning paradigm on RF-Agent, we remove self-verify and thought-align in Fig 6.c. The results show that the reasoning paradigm helps mitigate LLM hallucinations and provides effective evaluations, benefiting RF-Agent in generation and search. Furthermore, we also conduct an ablation study on the thought alignment component itself. Please refer to the AppendixF.6.

## 6  Conclusion

This paper proposes RF-Agent, which automates the design of high-performance reward functions for low-level control tasks by combining language agents and tree search. By treating the reward function design process as a decision problem, RF-Agent integrates MCTS into the reward function design process, utilizing the multi-stage contextual reasoning of LLMs to improve search efficiency and leveraging diverse action design with historical information to make effective improvements to the reward function. The results demonstrate that RF-Agent can effectively generate high-performance reward functions for various and complex low-level control tasks, validating its effectiveness.

**Limitation.** Our RF-Agent, similar to Eureka and Revolve, belongs to the category of reward function design methods that utilize feedback and LLM. These methods are rather computationally expensive and time-consuming due to requiring multiple interactions with the LLM and repeated policy training. While RF-Agent achieves better generation results with relatively smaller LLM models, it does not reduce the need for multiple RL training iterations. Our future work will focus on reducing the number of RL training cycles while maintaining the effective iterative improvement.

### Acknowledgments and Disclosure of Funding

This work was supported by the National Natural Science Foundation of China (Grant No.62031001, Grant No.62325101).

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

# A Environments

In this section,we provide environment details. For each environment,we list its observation and action dimensions,the verbatim task description,and the task evaluation score function $F$.

**IsaacGym.** The general objective of the IsaacGym[31] environments is to simulate various robotic tasks in a controlled virtual setting. These tasks range from balancing and controlling movements to interacting with objects. Each environment has specific goals such as making an ant run as fast as possible, having a humanoid perform tasks with agility, or controlling a hand to spin an object to a target orientation. The environments involve challenges in movement, manipulation, and stability, with each task requiring precise control of the robot or agent, measured by a corresponding evaluation score function that encourages the agent to optimize its performance.

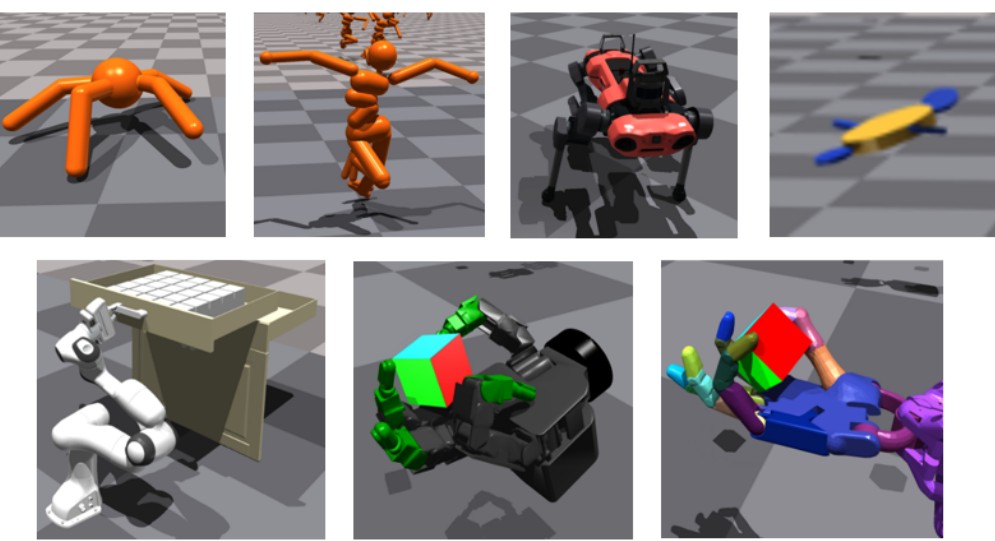

Figure 7: Examples of the IsaacGym environment.

| IsaacGym Environments |
| --- |
| Environment (obs dim, action dim)
Task description
Task evaluation score function $F$ |
| `Ant` (60,8)
To make the ant run forward as fast as possible
`cur_dist - prev_dist` |
| `Anymal` (48, 12)
To make the quadruped follow randomly chosen x, y, and yaw target velocities
`-(linvel_error + angvel_error)` |
| `Humanoid` (108, 21)
To make the humanoid run as fast as possible
`cur_dist - prev_dist` |
| `Quadcopter` (21, 12)
To make the quadcopter reach and hover near a fixed position
`-cur_dist` |
| `AllegroHand` (88, 16)
To make the hand spin the object to a target orientation |

```
number of consecutive successes where current success is 1[rot_dist <
0.1]
```

---

```
FrankaCabinet (23, 9)
```
To open the cabinet door
```
1[cabinet_pos > 0.39]
```

---

```
ShadowHand (211, 20)
```
To make the shadow hand spin the object to a target orientation
```
number of consecutive successes where current success is 1[rot_dist <
0.1]
```

---

**Bi-DexHands.** The Bi-DexHands[9] environment is designed to simulate dexterous manipulation tasks involving two hands. The overall goal of the Bi-DexHand environment is to enable the agent to perform various tasks that require coordinated manipulation using both hands. These tasks are modeled to simulate real-world dexterous activities, such as passing objects between hands, interacting with everyday tools, and performing precise actions requiring dual-hand cooperation. Tasks in Bi-DexHands environment are designed to push the boundaries of agent control, testing their ability to handle complex multi-step processes, balance, and precision, all of which are crucial in dexterous manipulation.

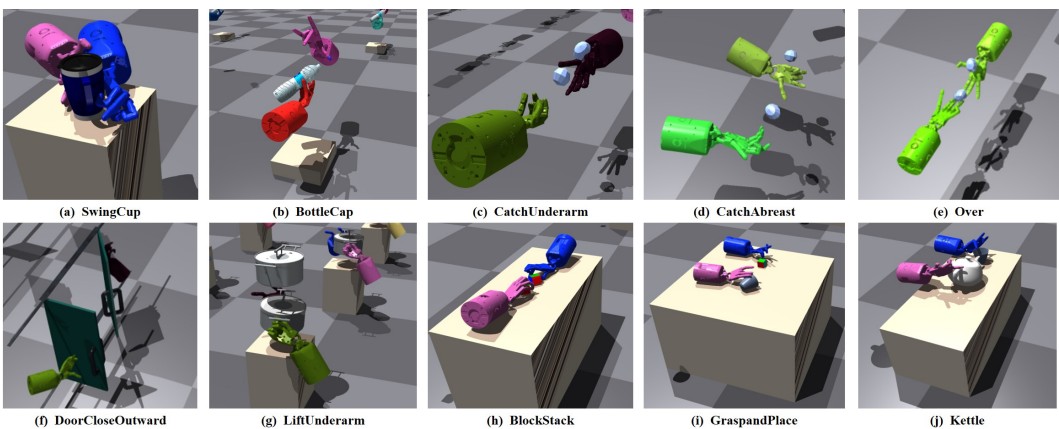

| (a) SwingCup | (b) BottleCap | (c) CatchUnderarm | (d) CatchAbreast | (e) Over |
| (f) DoorCloseOutward | (g) LiftUnderarm | (h) BlockStack | (i) GraspandPlace | (j) Kettle |

Figure 8: Examples of the Bi-DexHands environment.

---

| Bi-DexHands Environments |
| --- |

Environment (obs dim, action dim)
Task description
Task evaluation score function $F$

---

```
Over (398, 40)
```
This class corresponds to the HandOver task. This environment consists of two shadow hands with palms facing up, opposite each other, and an object that needs to be passed. In the beginning, the object will fall randomly in the area of the shadow hand on the right side. Then the hand holds the object and passes the object to the other hand. Note that the base of the hand is fixed. More importantly, the hand which holds the object initially can not directly touch the target, nor can it directly roll the object to the other hand, so the object must be thrown up and stays in the air in the process
```
1[dist < 0.03]
```

---

```
GraspAndPlace (425, 52)
```
This class corresponds to the GraspAndPlace task. This environment consists of dual-hands, an object and a bucket that requires us to pick up the object and put it into the bucket
```
1[|block- bucket| < 0.2]
```

---

`BottleCap` (420, 52)

This class corresponds to the Bottle Cap task. This environment involves two hands and a bottle, we need to hold the bottle with one hand and open the bottle cap with the other hand. This skill requires the cooperation of two hands to ensure that the cap does not fall

`1[dist > 0.03]`

---

`CatchUnderarm` (422, 52)

This class corresponds to the Catch Underarm task. In this task, two shadow hands with palms facing upwards are controlled to pass an object from one palm to the other. What makes it more difficult than the Hand over task is that the hands' translation and rotation degrees of freedom are no longer frozen but are added into the action space

`1[dist < 0.03]`

---

`BlockStack` (428, 52)

This class corresponds to the Block Stack task. This environment involves dual hands and two blocks, and we need to stack the block as a tower

`1[goal_dist_1 < 0.07 and goal_dist_2 < 0.07 and 50 * (0.05- z_dist_1) > 1]`

---

`SwingCup` (417, 52)

This class corresponds to the SwingCup task. This environment involves two hands and a dual handle cup, we need to use two hands to hold and swing the cup together

`1[rot_dist < 0.785]`

---

`LiftUnderarm` (417, 52)

This class corresponds to the LiftUnderarm task. This environment requires grasping the pot handle with two hands and lifting the pot to the designated position. This environment is designed to simulate the scene of lift in daily life and is a practical skill

`1[dist < 0.05]`

---

`Kettle` (417, 52)

This class corresponds to the PourWater task. This environment involves two hands, a kettle, and a bucket, we need to hold the kettle with one hand and the bucket with the other hand, and pour the water from the kettle into the bucket. In the practice task in Isaac Gym, we use many small balls to simulate the water

`1[|bucket- kettle_spout| < 0.05]`

---

`CatchAbreast` (422, 52)

This class corresponds to the Catch Abreast task. This environment consists of two shadow hands placed side by side in the same direction and an object that needs to be passed. Compared with the previous environment which is more like passing objects between the hands of two people, this environment is designed to simulate the two hands of the same person passing objects, so different catch techniques are also required and require more hand translation and rotation techniques

`1[dist] < 0.03`

---

`DoorCloseOutward` (417, 52)

This class corresponds to the DoorCloseOutward task. This environment also require a closed door to be opened and the door can only be pushed inward or initially open outward, but because they can't complete the task by simply pushing, which need to catch the handle by hand and then open or close it, so it is relatively difficult

`1[door_handle_dist < 0.5]`

---

# B  RF-Agent Details

## B.1  Basic Prompts.

Here, we present the fundamental prompts used across all LLM-based reward function design methods, including the system prompt, environment information prompt, code format tip prompt.

The **system prompt** is as follows:

You are a reward engineer trying to write reward functions to solve reinforcement learning tasks as effective as possible.

Your goal is to write a reward function for the environment that will help the agent learn the task described in text.

Your reward function should use useful variables from the environment as inputs.
As an example, the reward function signature can be:

```
@torch.jit.script
def compute_reward(object_pos: torch.Tensor, goal_pos: torch.Tensor) -> Tuple[
    torch.Tensor, Dict[str, torch.Tensor]]:
    ...
    return reward, {}
```

Since the reward function will be decorated with @torch.jit.script, please make sure that the code is compatible with TorchScript (e.g., use torch tensor instead of numpy array).

Make sure any new tensor or variable you introduce is on the same device as the input tensors.

The **environment information prompt** with task **IsaacGym-Ant** is as follows:

The task is: Ant. You need to make the ant run forward as fast as possible.

The Python environment is:

```
class Ant(VecTask):
    """Rest of the environment definition omitted."""
    def compute_observations(self):
        self.gym.refresh_dof_state_tensor(self.sim)
        self.gym.refresh_actor_root_state_tensor(self.sim)
        self.gym.refresh_force_sensor_tensor(self.sim)

        self.obs_buf[:], self.potentials[:], self.prev_potentials[:], self.up_vec
            [:], self.heading_vec[:] = compute_ant_observations(
            self.obs_buf, self.root_states, self.targets, self.potentials,
            self.inv_start_rot, self.dof_pos, self.dof_vel,
            self.dof_limits_lower, self.dof_limits_upper, self.dof_vel_scale,
            self.vec_sensor_tensor, self.actions, self.dt, self.
                contact_force_scale,
            self.basis_vec0, self.basis_vec1, self.up_axis_idx)

def compute_ant_observations(obs_buf, root_states, targets, potentials,
                             inv_start_rot, dof_pos, dof_vel,
                             dof_limits_lower, dof_limits_upper, dof_vel_scale,
                             sensor_force_torques, actions, dt,
                                 contact_force_scale,
                             basis_vec0, basis_vec1, up_axis_idx):
    # type: (Tensor, Tensor, Tensor, Tensor, Tensor, Tensor, Tensor, Tensor,
        Tensor, float, Tensor, Tensor, float, float, Tensor, Tensor, int) ->
        Tuple[Tensor, Tensor, Tensor, Tensor, Tensor]

    torso_position = root_states[:, 0:3]
    torso_rotation = root_states[:, 3:7]
    velocity = root_states[:, 7:10]
    ang_velocity = root_states[:, 10:13]

    to_target = targets - torso_position
    to_target[:, 2] = 0.0

    prev_potentials_new = potentials.clone()
    potentials = -torch.norm(to_target, p=2, dim=-1) / dt

    torso_quat, up_proj, heading_proj, up_vec, heading_vec =
        compute_heading_and_up(
        torso_rotation, inv_start_rot, to_target, basis_vec0, basis_vec1, 2)
```

```
        vel_loc, angvel_loc, roll, pitch, yaw, angle_to_target = compute_rot(
            torso_quat, velocity, ang_velocity, targets, torso_position)

        dof_pos_scaled = unscale(dof_pos, dof_limits_lower, dof_limits_upper)

        obs = torch.cat((torso_position[:, up_axis_idx].view(-1, 1), vel_loc,
            angvel_loc,
                        yaw.unsqueeze(-1), roll.unsqueeze(-1), angle_to_target.
                            unsqueeze(-1),
                        up_proj.unsqueeze(-1), heading_proj.unsqueeze(-1),
                            dof_pos_scaled,
                        dof_vel * dof_vel_scale, sensor_force_torques.view(-1, 24) *
                            contact_force_scale,
                        actions), dim=-1)

        return obs, potentials, prev_potentials_new, up_vec, heading_vec
```

The **environment information prompt** with task **BidexHands-Over** is as follows:

### Environment information Prompt with Task Bidex-Over

The task is: This class corresponds to the HandOver task. This environment consists of two shadow hands with palms facing up, opposite each other, and an object that needs to be passed. In the beginning, the object will fall randomly in the area of the shadow hand on the right side. Then the hand holds the object and passes the object to the other hand. Note that the base of the hand is fixed. More importantly, the hand which holds the object initially can not directly touch the target, nor can it directly roll the object to the other hand, so the object must be thrown up and stays in the air in the process.

The Python environment is:

```python
class ShadowHandOver(VecTask):
    """Rest of the environment definition omitted."""
    def compute_observations(self):
        self.gym.refresh_dof_state_tensor(self.sim)
        self.gym.refresh_actor_root_state_tensor(self.sim)
        self.gym.refresh_rigid_body_state_tensor(self.sim)
        self.gym.refresh_force_sensor_tensor(self.sim)
        self.gym.refresh_dof_force_tensor(self.sim)

        if self.obs_type in ["point_cloud"]:
            self.gym.render_all_camera_sensors(self.sim)
            self.gym.start_access_image_tensors(self.sim)

        self.object_pose = self.root_state_tensor[self.object_indices, 0:7]
        self.object_pos = self.root_state_tensor[self.object_indices, 0:3]
        self.object_rot = self.root_state_tensor[self.object_indices, 3:7]
        self.object_linvel = self.root_state_tensor[self.object_indices, 7:10]
        self.object_angvel = self.root_state_tensor[self.object_indices, 10:13]

        self.goal_pose = self.goal_states[:, 0:7]
        self.goal_pos = self.goal_states[:, 0:3]
        self.goal_rot = self.goal_states[:, 3:7]

        self.fingertip_state = self.rigid_body_states[:, self.fingertip_handles
            ][:, :, 0:13]
        self.fingertip_pos = self.rigid_body_states[:, self.fingertip_handles][:,
            :, 0:3]
        self.fingertip_another_state = self.rigid_body_states[:, self.
            fingertip_another_handles][:, :, 0:13]
        self.fingertip_another_pos = self.rigid_body_states[:, self.
            fingertip_another_handles][:, :, 0:3]

        if self.obs_type == "full_state":
            self.compute_full_state()
        elif self.obs_type == "point_cloud":
            self.compute_point_cloud_observation()

        if self.asymmetric_obs:
            self.compute_full_state(True)
```

The **code format tip prompt** is as follows:

**Code format tip Prompt**

Some helpful tips for writing the reward function code:

    (1) You may find it helpful to normalize to a fixed reward range like-1,1 or 0,1 by applying transformations like torch.exp() to the overall reward or its reward components in order to have reasonable values for the optimization procedure. This does not mean that you should bound the rewards as wrong bounds might hinder the learning algorithm from achieving the optimal policy.

    (2) If you choose to transform a reward component, then you must also introduce a temperature parameter inside the transformation function; this parameter must be a named variable in the reward function and it must not be an input variable. Each transformed reward component should have its own temperature variable.

    (3) If the task has a goal state, it might be helpful to add a bonus to the reward function if the agent achieves the goal (up to some tolerance).

    (4) Make sure the type of each input variable is correctly specified; a float input variable should not be specified as torch.Tensor.

    (5) Make sure you don't give contradictory reward components.

    (6) Most importantly, the reward code's input variables must contain only attributes of the provided environment class definition (namely, variables that have prefix self.). Under no circumstance can you introduce new input variables.

## B.2 Feedback Example and Analysis Prompts.

Here we use the Ant task as an example to show the specific content of feedback. This feedback is inherited from Eureka[28] and tracks the changes in a set of key variables during the training process, including the key components of the designed reward function, task scores and episode length.

**Feedback example on task Ant**

reward_forward_velocity:

    $'-0.02','1.07','1.47','1.90','2.29','2.62','3.00','3.48','3.54','3.67'$

, Max: 3.73, Mean: 2.48, Min: -0.02
reward_to_target:

    $'0.00','0.00','0.00','0.00','0.00','0.00','0.00','0.00','0.00','0.00'$

, Max: 0.00, Mean: 0.00, Min: 0.00
task_score:

    $'-0.02','1.08','1.48','1.90','2.29','2.60','2.97','3.45','3.49','3.62'$

, Max: 3.67, Mean: 2.46, Min: -0.02
episode_lengths:

$'59.19','180.39','285.69','425.37','519.28','578.28','634.42','644.16','633.70','649.94'$

, Max: 717.00, Mean: 494.84, Min: 59.19

Based on a given feedback, we provide some general tips for analyzing feedback.

**Trained results analysis tip**

(1) 'task_score' reflects the agent's actual task score or success rate after training under the current reward function.

> (2) If the values for a certain reward component are near identical throughout, then this means RL is not able to optimize this component as it is written.
> (3) If some reward components' magnitude is significantly larger or smaller, its value may not conducive to policy learning.

The above feedback mechanism will be used in the implementation of Revolve to achieve automatic feedback of revolve in the experimental environment.

### B.3 Prompts for RF-Agent Actions.

In RF-Agent, five different action types are set in the expansion stage, as well as the initial initialization action, as shown below:

The **initialization prompt** requires LLM to form a design idea based on the given environmental information and task description, and generate a reward function based on this idea.

> **Initialization prompt**
>
> First, describe the design idea and main steps of your reward function in one sentence. The description must be inside a brace outside the code implementation.
>
> Next, write a reward function based on this idea. The output of the reward function should consist of two items:
>    (1) the total reward,
>    (2) a dictionary of each individual reward component.
>
> The code output should be formatted as a python code string: "```python ... ```".
> Do not give additional explanations.

The **mutation** $a_{m1}$ **prompt** requires the LLM to locally optimize its reward function based on the feedback information of the parent node. In particular, $a_{m1}$ guides the LLM to modify the reward function composition mechanism.

> **Mutation $a_{m1}$ prompt**
>
> I have one reward function with its design idea and code as follows.
>
> Design Idea: {design_idea}
> Code: {reward_function}
>
> We trained a RL policy using the provided reward function code and tracked the values of the individual components in the reward function as well as global policy metrics such as success rates and episode lengths after every {epoch_freq} epochs and the maximum, mean, minimum values encountered:
>
> {trained_results}
> Analysis tips for trained results:
> {trained_result_analysis_tip}
>
> Please create a new reward function that has a different form but can be a modified version of the provided reward function. The new reward function should have a higher task score. Try to introduce more novel idea and add or remove some basic reward components from the environment.

The **mutation** $a_{m2}$ **prompt** requires the LLM to locally optimize its reward function based on the feedback information of the parent node. In particular, $a_{m2}$ guides the LLM to adjust the parameter weights within the reward function.

### Mutation $a_{m2}$ prompt

I have one reward function with its design idea and code as follows.

Design Idea: {design_idea}
Code: {reward_function}

We trained a RL policy using the provided reward function code and tracked the values of the individual components in the reward function as well as global policy metrics such as success rates and episode lengths after every {epoch_freq} epochs and the maximum, mean, minimum values encountered:

{trained_results}
Analysis tips for trained results:
{trained_result_analysis_tip}

Please identify the reward function parameters and create a new reward function that has a different parameter settings compared to the provided version. The new reward function should have a higher task score.

The **crossover** $a_{c3}$ **prompt** uses parent node information and nodes information in the elite set to generate a new reward function by combining the reward component. After sorting the scores corresponding to nodes in the set, the sampling of elite sets is randomly selected using the reciprocal of sorting as the weight.

### Crossover $a_{c3}$ prompt

I have {nums} existing reward functions with their design ideas and codes as follows.

{reward_func_group}

We trained a RL policy using the provided reward function code and tracked the values of the individual components in the reward function as well as global policy metrics such as success rates and episode lengths after every {epoch_freq} epochs and the maximum, mean, minimum values encountered:

{trained_results}
Analysis tips for trained results:
{trained_result_analysis_tip}

Please create a new reward function inspired by those reward functions. Try to list the common ideas in those high score reward functions and combine high-performing reward components from different reward functions based on the data tracked. The new reward function should have a higher task score.

The **path reasoning** $a_{r4}$ **prompt** uses parent node information and its ancestor information to generate a new reward function by reasoning this optimization path.

### Path Reasoning $a_{r4}$ prompt

I have {nums} existing reward functions related to optimization in sequence with their design ideas and codes as follows.

{reward_func_group}

We trained a RL policy using the provided reward function code and tracked the values of the individual components in the reward function as well as global policy metrics such as

> success rates and episode lengths after every {epoch_freq} epochs and the maximum, mean, minimum values encountered:
>
> {trained_results}
> Analysis tips for trained results:
> {trained_result_analysis_tip}
>
> Please create a new reward function inspired by all the above reward functions. Try to reason the optimization path and list some ideas in those reward functions that are clearly helpful to a better improvement. The new reward function should have a higher task score than any of them.

The **different thought** $a_{d5}$ **prompt** uses parent node information and random nodes information from the tree to generate a new reward function by requiring to design a new reward function with different structures from these nodes.

> **Different Thought $a_{d5}$ prompt**
>
> I have {nums} existing reward functions with their design ideas and codes as follows.
>
> {reward_func_group}
>
> We trained a RL policy using the provided reward function code and tracked the values of the individual components in the reward function as well as global policy metrics such as success rates and episode lengths after every {epoch_freq} epochs and the maximum, mean, minimum values encountered:
>
> {trained_results}
> Analysis tips for trained results:
> {trained_result_analysis_tip}
>
> Please create a new reward function that has a totally different form from the given algorithms. Try generating codes with different structures, flows or algorithms. Remember the new reward function should have a higher task score.

## B.4 Prompts for Thought-align and Self-verify.

**Thought-align.** After the reward function is successfully executed, the RF-Agent resummaries the reward function to eliminate the inconsistency between the original design idea and the reward function expression caused by hallucinations or modifications.

> **Thought-align prompt**
>
> Following is the Design Idea of a reward function for the problem and the code for implementing the reward function.
>
> Design Idea: {design_idea}
> Code: {reward_function}
>
> The content of the Design Idea cannot fully represent what the reward function has done informative. So, now you should re-describe the reward function using less than 4 sentences. Hint: You should reference the given Design Idea and highlight the most critical design ideas of the code. You can analyze the code to describe which reward components it contains, which observations they are calculated based on, what are the parameters and structure of the reward coupling process, and what special ideas are used.

**Self-verify.** In order to identify potential functions faster in the early selection stage, RF-Agent appropriately added a self-verify score from the LLM as part of the selection criteria, and the impact of this score will gradually weaken in the later stage.

> **Self-verify prompt**
>
> Be sure to remember the definition of the task. Additionally, I have an existing reward function with its design idea and code as follows.
>
> Design Idea: {design_idea}
> Code: {reward_function}
>
> Now, based on the task definition, imagine how an expert-level strategy completes the task, describe the execution planning and motion process of the expert strategy for the task, and then evaluate the given reward functions according to your imagined description process, and judge the similarity of the given reward function with the imagined process, with the numerical value limited to [-1,1].
> Finally, return the value enclosed in square brackets, such as [0.5].

## C    Training Details

**Hyper-parameters.** As shown in the paper, we deploy our RF-Agent with leaf node parallel scheme[7] in the expansion stage, we set this parallel number to 8. Thus, we configure the actions as $[2, 2, 2, 1, 1]$ per expansion to make the ratio of nodes that utilize local and global information at $1 : 1$. The initial value of $\lambda$ is set to 0.4 across all tasks, $v_{self}$ is constrained within the range of $[-1, 1]$ in order to have a more obvious distinction after softmax, $k$ is randomly sampled from $[2, 4]$ to control the number of nodes sampled in $a_{c3}$, $a_{r4}$ and $a_{d5}$, and $\eta$ is set to 0.7.

**Deployment resources.** We deployed RF-Agent on a 4 Nvidia Geforce RTX3090 cards with 128 core CPUs and 256GiB memory server. On issacgym, the average time consumed by each different task was 9 hours; on Bi-DexHands, the average time consumed by each task was 40 hours.

**Other evaluation details.** During the evaluation, we tested on 5 seeds different from those during the search. The number of training steps in the environment was consistent with the default training steps in the IssacGym and Bi-DexHands environments. For example, the uniform number of steps on Bi-DexHands is 1e8.

## D    Baseline Details

**Eureka.** Eureka[28] has been tested on IsaacGym and Bi-Dexhands using the GPT-4 model as the reward function generator. Given the high cost of GPT-4 and its reduced relevance as a mainstream choice, we used 4o and 4o-mini models for our experiments. Therefore, we also re-conducted experiments with Eureka using these two models. The parameter configuration for Eureka remains consistent with its default settings, maintaining 16 samples per iteration.

**Revolve.** Since Revolve[17] has not been previously tested on IsaacGym and Bi-Dexhands, we re-implemented the experiments. We retained its algorithm and evolutionary operation prompts, with the main changes being in the system prompt for the environment and replacing feedback information with automated environment feedback. These two types of prompts are consistent with those used in Eureka and RF-Agent. The parameter configuration for Revolve remains consistent with its default settings, maintaining 16 samples per iteration during population evolution.

## E    Cost Comparison

**Training resource consumption.** Eureka, Revolve and Ours RF-Agent are using the same total number of reward functions to training the policy, thus the training cost is almost the same.

**Storage Usage.** RF-Agent needs to store historical information of the entire tree, Revolve needs to store historical information of the population, and Eureka hardly needs to store historical information, but these additional historical information is composed of strings, whose order of magnitude is at the MB level.

**Token consumption.** The main consumption of each method is concentrated on the process of generating reward functions in combination with environmental information and instructions. RF-Agent also has thought-align and self-verify processes compared to Eureka and Revolve, but the token used in these two processes is much lower than the reward function generation process.

We tallied the input_token (prompt), output_token (completion), and the actual monetary cost for RF-Agent, Eureka, and Revolve on the Ant and Humanoid tasks. It is important to note that input_tokens can be processed in a single, efficient forward pass (a 'prefill'), whereas output_tokens are generated autoregressively (a 'decode' process), which is more computationally intensive. This efficiency difference is reflected in API pricing (e.g., Input: $0.6 per 1M tokens vs. Output: $2.4 per 1M tokens).

Table 4: Cost on Ant Environment

| Ant | Prefill Tokens | Decode Tokens | Total Cost ($) |
|---|---|---|---|
| Eureka | 82458 | 54505 | 0.1802868 |
| Revolve | 162348 | 43973 | 0.202944 |
| **Ours-total** | **257564** | **69704** | **0.321828** |
| Ours-action | 203282 | 53462 | - |
| Ours-others | 54282 | 16242 | - |

Table 5: Cost on Humanoid Environment

| Humanoid | Prefill Tokens | Decode Tokens | Total Cost ($) |
|---|---|---|---|
| Eureka | 78940 | 60021 | 0.191414 |
| Revolve | 169545 | 44118 | 0.20761 |
| **Ours-total** | **254555** | **66557** | **0.31247** |
| Ours-action | 204283 | 50270 | - |
| Ours-others | 50272 | 16287 | - |

The data in the tables leads to the following conclusions: RF-Agent utilizes 2-3 times more prefill tokens than the baselines, while the number of decode tokens is only moderately higher. Furthermore, we find that this asymmetric token consumption aligns perfectly with the design philosophy of RF-Agent. The higher usage of prefill tokens is a direct result of leveraging historical information (i.e., previously generated thoughts and reward functions). As the MCTS tree grows, these artifacts are flexibly combined and included in the prompt, forming a multi-step reasoning process for the LLM. This process enables the LLM to generate more sophisticated and effective reward functions, which in turn leads to the significant performance growth observed in our experiments.

## F  Additional Results

### F.1  Training Curves in Bi-DexHands.

In Fig.9, we show all the training curve results of the Expert-Easy and Expert-Hard groups on the selected Bi-DexHands. RF-Agent can maintain its dominant performance in most tasks.

### F.2  Reward Function Optimization Performance.

In Fig.10, we show the maximum score obtained on each task as the number of reward function samples increases. The results show that our RF-Agent has better iterative optimization capabilities on most tasks.

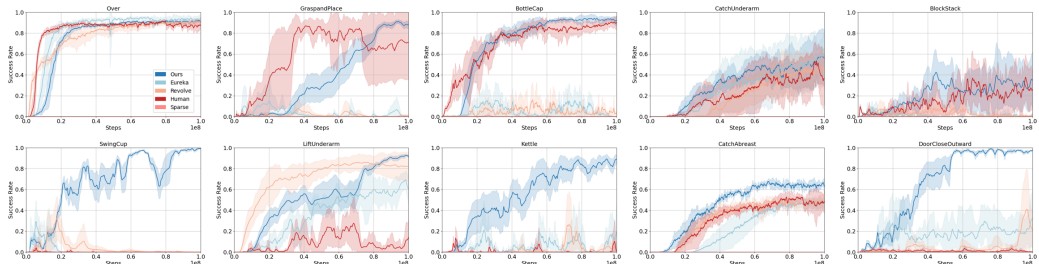

Figure 9: Success rates with exploration steps under reward functions by different methods on Bidex-hands.

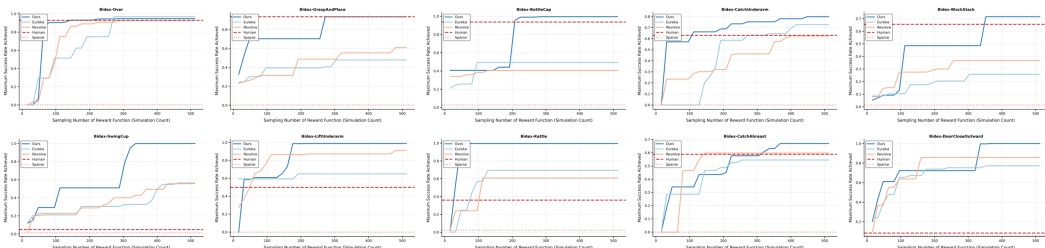

Figure 10: Reward function optimization performance with sampling counts in Bi-DexHands.

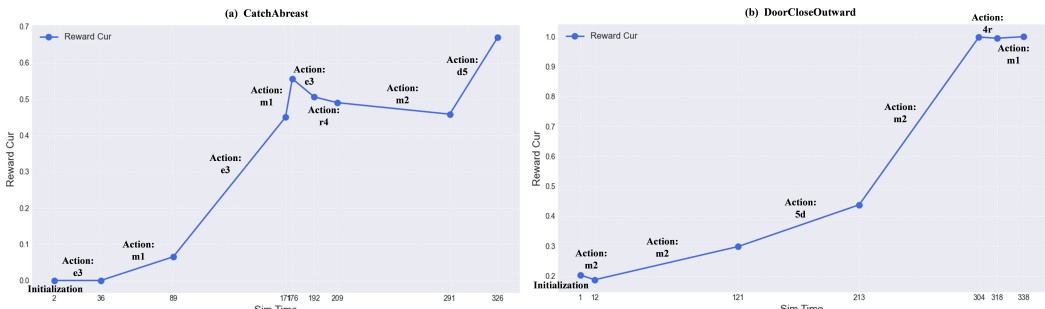

Figure 11: Examples of the best node growth path.

## F.3 More Optimization Path Examples of RF-Agent

We show the optimization paths of the nodes that end up with the best performance on two Bi-DexHands tasks, as shown in Fig.11. We found that the optimization paths on both tasks contain certain global actions, which shows that it is important to optimize reward functions using global historical information. In addition, the optimization paths on CatchAbreast further illustrate that those weak reward functions should not be easily abandoned, and mixing them with reward functions on other paths can also obtain performance advantage nodes.

## F.4 Out-of-distribution tasks performance

Considering that the training data for advanced large language model baselines might include certain common reward functions, we design and conduct two entirely new experiments using the Isaac Gym Ant environment:

**Ant Lie Down** Task description: "To make the ant lie down on its back (belly-up) as quickly and stably as possible. Its four feet should point towards the sky." The evaluation score for this task is based on a success condition, defined by an indicator function $1[\texttt{orientation\_reward} > 0.9]$, corresponding to a stable belly-up pose.

**Ant Patrol and Return** Task description: "To make the ant first run forward to the checkpoint position, then turn around and return to its starting origin position." For this task, success is $1[\texttt{task\_phase} == 1 \text{ and } \texttt{dist\_to\_start} < 0.5]$, signifying the completion of the entire two-stage trip.

Using the same experimental configuration as in our main Isaac Gym experiments and employing the GPT-4o-mini model, we obtained the following results:

Table 6: Comparison of methods on OOD-Ant tasks.

| Task | Sparse | Eureka | Revolve | Ours |
|---|---|---|---|---|
| Lie Down | 0.12±0.03 | 0.64±0.08 | 0.59±0.11 | 0.73±0.08 |
| Patrol and Return | 0.05±0.02 | 0.52±0.07 | 0.63±0.05 | 0.82±0.06 |

In these experiments, the 'Sparse' condition refers to using only the evaluation score (i.e., the 1[...] indicator function) as the reward signal. While it is not feasible to provide a handcrafted expert baseline for such novel tasks, the experimental results sufficiently demonstrate two key points: (1) The paradigm of LLM-based reward generation is capable of generalizing to these specialized, novel tasks. (2) Our RF-Agent continues to outperform the existing baseline methods on these new challenges.

### F.5 Detailed ablation studies on action combination

We further conducted a combinatorial study of action on Ant and Allegro Hand tasks:

Table 7: Detailed ablation study on action combinations.

| Action Combination | Ant | Allegro Hand |
|---|---|---|
| a w/ mutation | 5.13(-18%) | 18.76(-16%) |
| b w/ mutation + crossover | 5.77(-7%) | 21.90(-3%) |
| c w/ mutation + reasoning | 5.94(-4%) | 21.26(-5%) |
| d w/ crossover + reasoning | 4.85(-22%) | 19.21(-15%) |
| e w/ mutation + crossover + reasoning | 6.08(-2%) | 22.19(-2%) |
| **Full Version (w/ all actions)** | **6.22** | **22.52** |

By analyzing the ablation results, we can categorize our actions from the perspective of information utilization. The mutation action can be seen as a local operator, as it modifies a single parent node. In contrast, crossover and reasoning act as global operators, as they synthesize information from multiple nodes or entire ancestral paths.

The results clearly indicate that relying exclusively on one type of operator leads to a significant performance drop. For instance, using only a local action (w/ mutation) or only global actions (w/ crossover + reasoning) both result in considerable performance degradation. Therefore, a balanced combination of both local and global operators is crucial for achieving the best performance.

### F.6 Detailed ablation studies on thought align

We present an ablation study on the thought align mechanism. In our current design, this process consists of two steps:

(1) First Design Thought: Following a Chain-of-Thought like process, the LLM first generates a high-level 'design thought' before composing the actual reward function code.

(2) Second Thought Align: To mitigate potential hallucinations from a single generation step, after the reward function is generated, we prompt the LLM to re-examine the code's content and then revise the original 'design thought' to ensure it accurately reflects the code's logic.

To quantitatively validate the effectiveness of this mechanism, we conducted further ablations on these intermediate components:

The results indicate that the design thought mechanism is crucial across all tasks. Notably, the second thought align step has a particularly significant impact on the more complex manipulation task (Allegro Hand).

Table 8: Ablation study on thought alignment components.

| Task | w/ full thought align | w/o first design thought | w/o second thought align | w/o all |
|---|---|---|---|---|
| ant | 6.22 | 5.08(-18%) | 5.94(-4%) | 4.96(-20%) |
| allegro hand | 22.52 | 18.07(-19%) | 18.55(-17%) | 14.64(-35%) |

# G  Algorithm of RF-Agent

Algorithm 1 provides a pseudo-code for the proposed RF-Agent method, which can be combined with Fig.2 to further familiarize yourself with the entire RF-Agent process.

---

**Algorithm 1:** Reward Function Design via Language Agent Tree Search (RF-Agent)

---

**Input:** score Function $F$, reward designer $G$, the number of initial nodes $N_I$, max evaluation times $N$, action set $[a_{m1}, a_{m2}, a_{c3}, a_{r4}, a_{d5}]$, counts of action $[n_{m1}, n_{m2}, n_{c3}, n_{d4}, n_{d5}]$, UCT initial balance parameter $\lambda_0$

1  **// Initialization Stage**
2  Initialize a virtual root node $n_{root}$
3  Set $t \leftarrow 0$
4  Initialization $N_I$ nodes with designer $G$ and link all the $N_I$ nodes to the $n_{root}$ ;          // Eq.1
5  Evaluating $F$ after training policy $\pi$ with $R$ from $N_I$ nodes
6  **// MCTS Search**
7  **for** $t \leq N$ **do**
8      $\lambda \leftarrow \lambda_0 \cdot \frac{N-t}{N}$
9      **// MCTS Selection Stage**
10     $s \leftarrow n_{root}$
11     **while** $s$ is not a leaf node **do**
12        $s \leftarrow \text{argmax}_{s' \in \text{Child}(s)} \text{UCT}(s')$ ;                          // Eq.2 with $\lambda$
13     **end while**
14     **// MCTS Expansion Stage**
15     **for** $a$ *in action set* **do**
16        expand node $s$ with action $a$ and designer $G$ in $n_a$ times ;       // Eq.1 with $l_{action}$
17     **end**
18     **// MCTS Simulation Stage**
19     Evaluating $F$ after training policy $\pi$ with $R$ from newly generated nodes
20     **// MCTS Backpropagation Stage**
21     Evaluate the self-verify score $v_{self}$ of newly generated nodes.
22     **while** $s$ is not the root node $n_{root}$ **do**
23        Update $Q(s)$ and $N(s)$ ;                          // Eq.3
24     **end while**
25     $t \leftarrow t + \sum_{a \in \text{action set}} n_a$
26 **end**
27 **Return:** The node $s^*$ with the best reward function $R^*$

---

# H  Licenses

The licenses and URL of baselines , benchmarks and deployment algorithm are listed in 9.

Table 9: The licenses and URL of baselines , benchmarks and deployment algorithm.

| Resources | Type | License | URL |
|---|---|---|---|
| Eureka[28] | Codes for Baseline | MIT | `https://github.com/eureka-research/Eureka` |
| Revolve[17] | Codes for Baseline | Available Online | `https://github.com/RishiHazra/Revolve/tree/main` |
| IsaacGym[31] | Codes for Benchmark | BSD 3-Clause License | `https://github.com/isaac-sim/IsaacGymEnvs` |
| Bi-DexHands[9] | Codes for Benchmark | Apache 2.0 | `https://github.com/PKU-MARL/DexterousHands` |
| RL-games[30] | Codes for Algorithm | MIT | `https://github.com/Denys88/rl_games` |

# I RF-Agent Reward Examples

## I.1 Reward after Diverse Actions Examples

We randomly selected a task and a corresponding node to show what the tendency of the new reward function generated by RF-Agent under different actions. Here we select the node with RF-Agent developed on CatchAbreast to the path "root-0i-e3-m1-e3" as the current node. First, the reward function of the node is displayed, and then the reward function generated by different actions under the node is displayed in turn.

Current reward function on CatchAbreast.

```python
@torch.jit.script
def compute_reward(object_pos: torch.Tensor, goal_pos: torch.Tensor,
    left_hand_pos: torch.Tensor, right_hand_pos: torch.Tensor) -> Tuple[torch.
    Tensor, Dict[str, torch.Tensor]]:
    distance_to_goal = torch.norm(object_pos - goal_pos, dim=-1)
    distance_to_left_hand = torch.norm(left_hand_pos - object_pos, dim=-1)
    distance_to_right_hand = torch.norm(right_hand_pos - object_pos, dim=-1)

    # Reward for being close to the goal
    distance_reward = torch.clamp(1.0 - (distance_to_goal / 0.5), min=0, max=1)

    # Reward for successfully passing the object, bonus for being within a
        proximity threshold
    success_bonus = torch.where(distance_to_goal < 0.1, torch.tensor(1.0, device=
        object_pos.device), torch.tensor(0.0, device=object_pos.device))

    # Hand balance reward penalizing uneven movement
    hand_balance_reward = -torch.abs(distance_to_left_hand -
        distance_to_right_hand)

    # Proximity reward to encourage hands to efficiently reach the object
    combined_hand_distance = distance_to_left_hand + distance_to_right_hand
    proximity_reward = torch.clamp(1.0 - (combined_hand_distance / 0.8), min=0,
        max=1)

    # Smoothness penalty to encourage fluid hand movements
    hand_distance_between_hands = torch.norm(left_hand_pos - right_hand_pos, dim
        =-1)
    smooth_movement_penalty = 0.01 * hand_distance_between_hands

    # Total reward calculation
    total_reward = distance_reward + success_bonus + hand_balance_reward +
        proximity_reward - smooth_movement_penalty

    # Reward components dictionary
    reward_components = {
        "distance_reward": distance_reward,
        "success_bonus": success_bonus,
        "hand_balance_reward": hand_balance_reward,
        "proximity_reward": proximity_reward,
        "smooth_movement_penalty": smooth_movement_penalty
    }

    return total_reward, reward_components
```

**Mutation $a_{m1}$.**

Reward function after $a_{m1}$ on CatchAbreast.

```python
@torch.jit.script
def compute_reward(object_pos: torch.Tensor, goal_pos: torch.Tensor,
    left_hand_pos: torch.Tensor, right_hand_pos: torch.Tensor) -> Tuple[torch.
    Tensor, Dict[str, torch.Tensor]]:
    distance_to_goal = torch.norm(object_pos - goal_pos, dim=-1)
    distance_to_left_hand = torch.norm(left_hand_pos - object_pos, dim=-1)
    distance_to_right_hand = torch.norm(right_hand_pos - object_pos, dim=-1)

    # Reward for being very close to the goal with increased distance tolerance
    distance_reward = torch.clamp(1.0 - (distance_to_goal / 0.6), min=0, max=1)
```

```python
        # Success bonus increased for being within a tighter proximity threshold
        success_bonus = torch.where(distance_to_goal < 0.05, torch.tensor(1.0, device
            =object_pos.device), torch.tensor(0.0, device=object_pos.device))

        # Balance reward penalizing uneven movement but normalizing based on hand
            distance
        hand_balance_penalty = -torch.abs(distance_to_left_hand -
            distance_to_right_hand)
        normalized_hand_distance = torch.clamp((distance_to_left_hand +
            distance_to_right_hand) / (0.6 + 0.6), min=0, max=1)
        hand_balance_reward = hand_balance_penalty * normalized_hand_distance

        # Proximity reward encouraging efficient hand placement
        combined_hand_distance = distance_to_left_hand + distance_to_right_hand
        proximity_reward = torch.clamp(1.0 - (combined_hand_distance / 0.75), min=0,
            max=1)

        # Smoothness bonus for transitions encouraging fluid hand movements
        hand_distance_between_hands = torch.norm(left_hand_pos - right_hand_pos, dim
            =-1)
        smooth_movement_bonus = torch.clamp(1.0 - (hand_distance_between_hands / 0.1)
            , min=0, max=1)

        # Total reward calculation
        total_reward = (distance_reward + success_bonus + hand_balance_reward +
            proximity_reward + smooth_movement_bonus) / 1.5

        # Reward components dictionary
        reward_components = {
            "distance_reward": distance_reward,
            "success_bonus": success_bonus,
            "hand_balance_reward": hand_balance_reward,
            "proximity_reward": proximity_reward,
            "smooth_movement_bonus": smooth_movement_bonus
        }

        return total_reward, reward_components
```

where the key change is adding the normalizing mechanism to "hand_balance_penalty"

**Mutation** $a_{m2}$.

Reward function after $a_{m2}$ on CatchAbreast.

```python
@torch.jit.script
def compute_reward(object_pos: torch.Tensor, goal_pos: torch.Tensor,
    left_hand_pos: torch.Tensor, right_hand_pos: torch.Tensor) -> Tuple[torch.
    Tensor, Dict[str, torch.Tensor]]:
    distance_to_goal = torch.norm(object_pos - goal_pos, dim=-1)
    distance_to_left_hand = torch.norm(left_hand_pos - object_pos, dim=-1)
    distance_to_right_hand = torch.norm(right_hand_pos - object_pos, dim=-1)

    # Reward for being close to the goal with a larger range
    distance_reward = torch.clamp(1.0 - (distance_to_goal / 0.3), min=0, max=1)

    # Larger success bonus for passing the object
    success_bonus = torch.where(distance_to_goal < 0.05, torch.tensor(2.0, device
        =object_pos.device), torch.tensor(0.0, device=object_pos.device))

    # Hand balance reward maintaining a moderate penalty
    hand_balance_reward = -0.5 * torch.abs(distance_to_left_hand -
        distance_to_right_hand)

    # Enhanced proximity reward encouraging efficient hand placement
    combined_hand_distance = distance_to_left_hand + distance_to_right_hand
    proximity_reward = torch.clamp(1.0 - (combined_hand_distance / 0.6), min=0,
        max=1)

    # Smoothness penalty slightly adjusted to minimize harsh penalties
    hand_distance_between_hands = torch.norm(left_hand_pos - right_hand_pos, dim
        =-1)
    smooth_movement_penalty = 0.005 * hand_distance_between_hands
```

```
        # Total reward calculation
        total_reward = distance_reward + success_bonus + hand_balance_reward +
            proximity_reward - smooth_movement_penalty

        # Reward components dictionary
        reward_components = {
            "distance_reward": distance_reward,
            "success_bonus": success_bonus,
            "hand_balance_reward": hand_balance_reward,
            "proximity_reward": proximity_reward,
            "smooth_movement_penalty": smooth_movement_penalty
        }

        return total_reward, reward_components
```

where the key change is change the parameters in some reward components.

**Crossover** $a_{c3}$.

Reward function after $a_{c3}$ on CatchAbreast.

```
@torch.jit.script
def compute_reward(object_pos: torch.Tensor, goal_pos: torch.Tensor,
    left_hand_pos: torch.Tensor, right_hand_pos: torch.Tensor) -> Tuple[torch.
    Tensor, Dict[str, torch.Tensor]]:
    distance_to_goal = torch.norm(object_pos - goal_pos, dim=1)
    distance_to_left_hand = torch.norm(left_hand_pos - object_pos, dim=1)
    distance_to_right_hand = torch.norm(right_hand_pos - object_pos, dim=1)

    goal_reward = torch.clamp(1.0 - (distance_to_goal / 0.5), min=0, max=1)  #
        Proximity to goal
    success_bonus = torch.where(distance_to_goal < 0.1, torch.tensor(2.0, device=
        object_pos.device), torch.tensor(0.0, device=object_pos.device))  #
        Success bonus

    hand_balance_reward = -torch.abs(distance_to_left_hand -
        distance_to_right_hand)  # Penalizes uneven movement between hands
    hand_coordination_reward = torch.exp(-torch.norm(left_hand_pos -
        right_hand_pos, dim=1))  # Encourages closer hand positions
    proximity_reward = -0.5 * (distance_to_left_hand + distance_to_right_hand)  #
         Maintain proximity to the object

    total_reward = goal_reward + success_bonus + hand_balance_reward +
        hand_coordination_reward + proximity_reward

    reward_components = {
        "goal_reward": goal_reward,
        "success_bonus": success_bonus,
        "hand_balance_reward": hand_balance_reward,
        "hand_coordination_reward": hand_coordination_reward,
        "proximity_reward": proximity_reward
    }

    return total_reward, reward_components
```

where the key change is to delete "combined_hand_distance" and "smooth_movement_penalty" from the initial reward function while keeping other high-performing components.

**Path Reasoning** $a_{r4}$.

Reward function after $a_{r4}$ on CatchAbreast.

```
@torch.jit.script
def compute_reward(object_pos: torch.Tensor, goal_pos: torch.Tensor,
    left_hand_pos: torch.Tensor, right_hand_pos: torch.Tensor) -> Tuple[torch.
    Tensor, Dict[str, torch.Tensor]]:
    # Constants
    distance_threshold = 0.1
    success_bonus = 1.0
    smooth_movement_penalty = 0.01
```

```
        # Distance metrics
        distance_to_goal = torch.norm(object_pos - goal_pos, dim=1)
        distance_to_left_hand = torch.norm(left_hand_pos - object_pos, dim=1)
        distance_to_right_hand = torch.norm(right_hand_pos - object_pos, dim=1)

        # Reward components
        distance_reward = torch.clamp(1.0 - (distance_to_goal / 0.5), min=0, max=1)
            # Reward for being close to the goal

        # Reward for successfully passing the object
        successful_pass = (distance_to_goal < distance_threshold).type(object_pos.
            dtype)
        reward_successful_pass = successful_pass * success_bonus

        # Hand balance reward to encourage synchronized movements
        hand_balance_reward = -torch.abs(distance_to_left_hand -
            distance_to_right_hand)

        # Proximity reward for effectively reaching the object
        combined_hand_distance = distance_to_left_hand + distance_to_right_hand
        proximity_reward = torch.clamp(1.0 - (combined_hand_distance / 0.8), min=0,
            max=1)

        # Smoothness penalty for fluid hand movements
        hand_distance_between_hands = torch.norm(left_hand_pos - right_hand_pos, dim
            =1)
        smooth_movement_penalty_reward = -smooth_movement_penalty *
            hand_distance_between_hands

        # Total reward calculation
        total_reward = distance_reward + reward_successful_pass + hand_balance_reward
             + proximity_reward + smooth_movement_penalty_reward

        # Reward components dictionary
        reward_components = {
            "distance_reward": distance_reward,
            "reward_successful_pass": reward_successful_pass,
            "hand_balance_reward": hand_balance_reward,
            "proximity_reward": proximity_reward,
            "smooth_movement_penalty_reward": smooth_movement_penalty_reward
        }

        return total_reward, reward_components
```

where the key change is that the coefficients in the original reward function have been redefined and some nested methods have been changed.

**Different Thought** $a_{d5}$.

Reward function after $a_{r4}$ on CatchAbreast.

```
@torch.jit.script
def compute_reward(object_pos: torch.Tensor, goal_pos: torch.Tensor,
    left_hand_pos: torch.Tensor, right_hand_pos: torch.Tensor, object_linvel:
    torch.Tensor) -> Tuple[torch.Tensor, Dict[str, torch.Tensor]]:
        # Design Idea: Utilize a nested reward structure focusing on progressive
            subtasks of grasping and passing the object, rewarding successful
            transitions and smooth trajectories.

        # Define parameters
        goal_tolerance = 0.1
        grasp_radius = 0.05
        smoothness_weight = 0.2

        # Calculate distances for nested rewards
        distance_to_goal = torch.norm(object_pos - goal_pos, dim=-1)
        distance_to_left_hand = torch.norm(object_pos - left_hand_pos, dim=-1)
        distance_to_right_hand = torch.norm(object_pos - right_hand_pos, dim=-1)

        # Nested reward for successful grasping (within grasp radius)
        grasp_reward = torch.where(distance_to_left_hand < grasp_radius, torch.tensor
            (0.5, device=object_pos.device), torch.zeros_like(distance_to_left_hand))
```

```
        grasp_reward += torch.where(distance_to_right_hand < grasp_radius, torch.
            tensor(0.5, device=object_pos.device), torch.zeros_like(
            distance_to_right_hand))

        # Goal reward
        goal_reward = torch.where(distance_to_goal < goal_tolerance, torch.tensor
            (1.0, device=object_pos.device), torch.zeros_like(distance_to_goal))

        # Smoothness reward calculated as the negative exponential of object linear
            velocity
        smoothness_reward = -torch.exp(-smoothness_weight * torch.norm(object_linvel,
            dim=-1))

        # Total reward combining all components
        total_reward = smoothness_reward + grasp_reward + goal_reward

        # Reward components
        reward_components = {
            "grasp_reward": grasp_reward,
            "goal_reward": goal_reward,
            "smoothness_reward": smoothness_reward
        }

        return total_reward, reward_components
```

where the change is to think about new reward forms, introducing a grasping angle and nesting it into the distance-related rewards.

## I.2   Design Thought after Alignment

Similarly, the current node "root-0i-e3-m1-e3" in the previous section is an example. The design thought during the generation process is:

> **Before thought-align**
>
> The reward function is designed to optimize agent performance in the Catch Abreast task by rewarding careful proximity of the hands to the object, effective synchronization of hand movements, and incentivizing reaching the goal with a success bonus, while penalizing excessive movement to ensure fluid hand coordination.

After thought alignment:

> **After thought-align**
>
> The reward function is designed to enhance agent performance in the Catch Abreast task by incorporating multiple components that prioritize hand coordination and goal achievement. It evaluates proximity to the goal and the object, providing rewards for closeness and a success bonus for passing the object within a defined threshold. Additionally, it incorporates a hand balance reward to penalize uneven movements between the hands, a proximity reward to encourage efficient hand placement, and a smoothness penalty to promote fluid motion. The total reward is calculated by summing these components while ensuring that fluid hand coordination is incentivized.

Obviously, the design ideas after thinking alignment can more comprehensively summarize the corresponding reward function.

## I.3   RF-Agent and Baselines Reward Comparison

Here we list some best reward function from different methods.

**IsaacGym-Ant.**

**Our RF-Agent.**

```python
@torch.jit.script
def compute_reward(root_states: torch.Tensor, heading_vec: torch.Tensor, up_vec:
    torch.Tensor) -> Tuple[torch.Tensor, Dict[str, torch.Tensor]]:
    # Extract the forward velocity in the x-direction
    velocity = root_states[:, 7:10]
    forward_velocity = velocity[:, 0]

    # Weight for different reward components
    heading_weight = 1.5
    upright_weight = 0.5
    y_z_penalty_weight = 0.3

    # Cosine similarity reward for maintaining the desired heading direction (x-
        axis)
    desired_heading = torch.tensor([1.0, 0.0, 0.0], device=heading_vec.device).
        expand_as(heading_vec)
    heading_cos_sim = torch.nn.functional.cosine_similarity(desired_heading,
        heading_vec, dim=-1)
    heading_reward = heading_weight * (heading_cos_sim * forward_velocity)

    # Stability penalty for velocity in y and z directions
    y_z_velocity_penalty = y_z_penalty_weight * torch.norm(velocity[:, 1:], p=2,
        dim=-1)

    # Reward for being upright, using the z-component of the up_vec to measure
        uprightness
    upright_bonus_temp = 2.0
    upright_bonus = upright_weight * torch.exp(upright_bonus_temp * (up_vec[:, 2]
        - 1.0))

    # Total reward is a combination of forward movement, heading alignment, and
        uprightness
    total_reward = forward_velocity + heading_reward + upright_bonus -
        y_z_velocity_penalty

    # Construct reward components dictionary
    reward_components = {
        "forward_velocity": forward_velocity,
        "heading_reward": heading_reward,
        "upright_bonus": upright_bonus,
        "y_z_velocity_penalty": -y_z_velocity_penalty
    }

    return total_reward, reward_components
```

**Eureka.**

```python
@torch.jit.script
def compute_reward(root_states: torch.Tensor, potentials: torch.Tensor,
    prev_potentials: torch.Tensor, actions: torch.Tensor,
                   dof_vel: torch.Tensor, dof_limits_lower: torch.Tensor,
                       dof_limits_upper: torch.Tensor) -> Tuple[torch.Tensor,
                       Dict[str, torch.Tensor]]:
    # Extract relevant state information
    velocity = root_states[:, 7:10]  # Linear velocity
    torso_position = root_states[:, 0:3]  # Position of the torso

    # Reward component to encourage forward velocity
    forward_velocity_reward = velocity[:, 0]  # Assuming forward is along the x-
        axis

    # Modify target closeness reward to reflect more sensitivity
    target_closeness_reward_temperate = 0.2
    target_closeness_reward = prev_potentials - potentials
    exp_target_closeness_reward = torch.exp(target_closeness_reward_temperate *
        target_closeness_reward)

    # Re-examine dof velocity penalty for more sensitivity
    dof_vel_penalty_temperate = 0.02
    dof_vel_penalty = torch.sum(torch.abs(dof_vel), dim=-1)
    exp_dof_vel_penalty = torch.exp(-dof_vel_penalty_temperate * dof_vel_penalty)
```

```
        # Actions penalty adjust temperature for greater distinction
        actions_penalty_temperate = 0.005
        actions_penalty = torch.sum(torch.abs(actions), dim=-1)
        exp_actions_penalty = torch.exp(-actions_penalty_temperate * actions_penalty)

        # Total normalized reward
        total_reward = forward_velocity_reward + 0.5 * exp_target_closeness_reward +
            0.1 * exp_dof_vel_penalty + 0.1 * exp_actions_penalty

        # Compile individual components into a dictionary
        reward_components = {
            "forward_velocity_reward": forward_velocity_reward,
            "exp_target_closeness_reward": exp_target_closeness_reward,
            "exp_dof_vel_penalty": exp_dof_vel_penalty,
            "exp_actions_penalty": exp_actions_penalty,
        }

        return total_reward, reward_components
```

## Revolve.

```
@torch.jit.script
def compute_reward(root_states: torch.Tensor, targets: torch.Tensor, potentials:
    torch.Tensor,
                   prev_potentials: torch.Tensor, up_vec: torch.Tensor) -> Tuple[
                       torch.Tensor, Dict[str, torch.Tensor]]:
    # Extract necessary components
    velocity = root_states[:, 7:10]

    # Forward speed reward component
    forward_speed_reward = velocity[:, 0]  # Forward speed along x-axis

    # Forward potential difference reward component
    forward_potential_diff = potentials - prev_potentials

    # Upright reward component
    up_vec_goal = torch.tensor([0.0, 0.0, 1.0], device=up_vec.device).unsqueeze
        (0).expand_as(up_vec)
    upright_reward = torch.sum(up_vec * up_vec_goal, dim=-1)

    # Set temperatures for exponential scaling
    forward_speed_temp = 0.35
    forward_potential_temp = 0.4
    upright_temp = 1.0  # Keeping original scale

    # Exponentially scale the rewards
    scaled_forward_speed_reward = forward_speed_temp * torch.exp(
        forward_speed_reward * forward_speed_temp)
    scaled_forward_potential_diff = forward_potential_temp * torch.exp(
        forward_potential_diff * forward_potential_temp)
    scaled_upright_reward = torch.exp(upright_temp * upright_reward)

    # Total reward calculation
    total_reward = scaled_forward_speed_reward + scaled_forward_potential_diff +
        scaled_upright_reward

    # Reward components dictionary
    reward_components = {
        "forward_speed_reward": scaled_forward_speed_reward,
        "forward_potential_diff_reward": scaled_forward_potential_diff,
        "upright_reward": scaled_upright_reward
    }

    return total_reward, reward_components
```

**Bidex-CatchAbreast.**

## Our RF-Agent.

```python
@torch.jit.script
def compute_reward(object_pos: torch.Tensor, goal_pos: torch.Tensor,
    left_hand_pos: torch.Tensor, right_hand_pos: torch.Tensor) -> Tuple[torch.
    Tensor, Dict[str, torch.Tensor]]:
    distance_to_goal = torch.norm(object_pos - goal_pos, dim=1)
    distance_to_left_hand = torch.norm(left_hand_pos - object_pos, dim=1)
    distance_to_right_hand = torch.norm(right_hand_pos - object_pos, dim=1)

    # Reward for proximity to the goal with increased tolerance
    distance_reward = torch.clamp(1.0 - (distance_to_goal / 0.5), min=0, max=1)

    # Success bonus for accurate object passing with tighter threshold
    success_bonus = torch.where(distance_to_goal < 0.03, torch.tensor(1.0, device
        =object_pos.device), torch.tensor(0.0, device=object_pos.device))

    # Hand balance reward incentivizing even distribution of effort between hands
    hand_balance_reward = -0.5 * torch.abs(distance_to_left_hand -
        distance_to_right_hand) * torch.clamp(1.0 - (distance_to_left_hand +
        distance_to_right_hand) / 1.5, min=0, max=1)

    # Proximity reward to encourage both hands to be near the object
    combined_hand_distance = distance_to_left_hand + distance_to_right_hand
    proximity_reward = torch.clamp(1.0 - (combined_hand_distance / 0.75), min=0,
        max=1)

    # Smoothness bonus to promote fluid hand movements
    hand_distance_between_hands = torch.norm(left_hand_pos - right_hand_pos, dim
        =1)
    smooth_movement_bonus = torch.clamp(1.0 - (hand_distance_between_hands / 0.1)
        , min=0, max=1)

    # Total reward calculation
    total_reward = (distance_reward + success_bonus + hand_balance_reward +
        proximity_reward + smooth_movement_bonus) / 2.0

    # Reward components dictionary
    reward_components = {
        "distance_reward": distance_reward,
        "success_bonus": success_bonus,
        "hand_balance_reward": hand_balance_reward,
        "proximity_reward": proximity_reward,
        "smooth_movement_bonus": smooth_movement_bonus
    }

    return total_reward, reward_components
```

## Eureka.

```python
@torch.jit.script
def compute_reward(object_pos: torch.Tensor, goal_pos: torch.Tensor,
                   object_linvel: torch.Tensor) -> Tuple[torch.Tensor, Dict[str,
                       torch.Tensor]]:

    # Constants for temperature adjustments
    distance_temp = 5.0  # Increased temperature sensitivity for distance
    velocity_temp = 1.0   # Reduced scaling for velocity

    # Compute distance to goal
    distance_to_goal = torch.norm(object_pos - goal_pos, dim=1)
    # Negative distance to promote minimization
    distance_reward = torch.exp(-distance_temp * distance_to_goal)  # Exponential
         decay for distance

    # Improved velocity reward with penalty for slow speeds
    speed_threshold = 0.1  # threshold for penalty
    velocity_magnitude = torch.norm(object_linvel, dim=1)
    # Penalty for low speeds to encourage proper movement
    velocity_penalty = torch.where(velocity_magnitude < speed_threshold, -1.0 * (
        speed_threshold - velocity_magnitude), torch.zeros_like(
        velocity_magnitude))

    # Positive reward for aligned velocity towards the goal direction
```

```
        direction_vector = goal_pos - object_pos  # Direction vector from object to
            goal
        direction_norm = torch.norm(direction_vector, dim=1, keepdim=True) + 1e-6
        desired_velocity = direction_vector / direction_norm  # Normalized direction
        aligned_velocity = torch.sum(object_linvel * desired_velocity, dim=1)  # Dot
            product for alignment
        aligned_velocity_reward = torch.clamp(aligned_velocity, min=0.0)  # Only
            reward positive velocities
        velocity_reward = aligned_velocity_reward + velocity_penalty  # Combine
            rewards and penalties

        # Normalize velocity reward
        velocity_reward = torch.clamp(velocity_reward, min=0.0)

        # Combine all rewards to form total reward
        total_reward = distance_reward + velocity_reward

        # Components for debugging
        reward_components = {
            'distance_reward': distance_reward,
            'velocity_reward': velocity_reward,
        }

        return total_reward, reward_components
```

## Revolve.

```
@torch.jit.script
def compute_reward(object_pos: torch.Tensor, goal_pos: torch.Tensor,
    left_hand_pos: torch.Tensor, right_hand_pos: torch.Tensor, left_hand_rot:
    torch.Tensor, right_hand_rot: torch.Tensor) -> Tuple[torch.Tensor, Dict[str,
    torch.Tensor]]:
    # Define temperature variables for normalization
    temp_pos = 0.1
    temp_rot = 0.05
    temp_catch = 0.2  # Adjusted temperature for catch reward

    # Compute the distance from the object to the goal
    distance_to_goal = torch.norm(object_pos - goal_pos, dim=-1)
    reward_position = -distance_to_goal  # Closer is better, so we take negative

    # Compute the distance from the object to the left and right hands
    distance_to_left_hand = torch.norm(object_pos - left_hand_pos, dim=-1) + 1e-6
            # Adding a small epsilon to prevent log(0)
    distance_to_right_hand = torch.norm(object_pos - right_hand_pos, dim=-1) + 1e
        -6

    # Encourage the object to be close to the hands with stronger feedback
    reward_catch_left_base = -torch.exp(distance_to_left_hand / temp_catch)  #
        Base feedback
    reward_catch_left_prox = torch.where(distance_to_left_hand < 0.1, 1.0 -
        distance_to_left_hand / 0.1, torch.tensor(0.0, device=object_pos.device))
            # Linear boost for proximity
    reward_catch_left = reward_catch_left_base + reward_catch_left_prox  #
        Combined rewards

    reward_catch_right = -1.5 * torch.exp(distance_to_right_hand / temp_catch)  #
        More weight for right hand penalty
    reward_catch = reward_catch_left + reward_catch_right  # Combined rewards for
        better guidance

    # Compute alignment/rotation rewards based on the orientation of the hands
    desired_left_rot = torch.atan2(left_hand_pos[:, 1], left_hand_pos[:, 0])
    desired_right_rot = torch.atan2(right_hand_pos[:, 1], right_hand_pos[:, 0])

    reward_left_rot = -torch.abs(left_hand_rot[:, 0] - desired_left_rot)  #
        Compare some angle derived from quaternion
    reward_right_rot = -torch.abs(right_hand_rot[:, 0] - desired_right_rot)

    # Combine all rewards
    total_reward = torch.exp(reward_position / temp_pos) + torch.exp(reward_catch
        / temp_catch) + torch.exp(reward_left_rot / temp_rot) + torch.exp(
        reward_right_rot / temp_rot)

    # Prepare individual rewards to return
```

```python
        reward_components = {
            'reward_position': reward_position,
            'reward_catch': reward_catch,
            'reward_catch_left': reward_catch_left,
            'reward_catch_right': reward_catch_right,
            'reward_left_rot': reward_left_rot,
            'reward_right_rot': reward_right_rot,
        }

        return total_reward, reward_components
```

