# OpenReview forum: "RF-Agent: Automated Reward Function Design via Language Agent Tree Search"
_NeurIPS.cc/2025/Conference — NeurIPS 2025 spotlight_

### Official Review · Reviewer_qnw7 · 2025-06-23

**Clarity:** 3
**Significance:** 2
**Originality:** 3
**Rating:** 5
**Confidence:** 4

**Summary:**

This paper proposes RF-Agent, an LLM-based framework for automated reward function design using  Monte Carlo Tree Search.
It treats reward function design as a sequence of mutations, crossovers, etc., and uses LLM to modify the Python code for the reward function, allowing for iterative refinement based on training feedback, task descriptions, and generated design ideas.
Experiments on IsaacGym and Bi-DexHands benchmarks show improved performance over prior LLM-based baselines (Eureka, Revolve).

**Questions:**

- l. 50--52: Is there evidence or references showing it's because of exploration and exploitation, not other reasons?
- l. 112: $\mathcal{R}$ is the set of $[S \times A, \mathbb{R}]$ of all real-valued functions over $S \times A$, right?
- l. 113: I understand $\pi: S \to \Delta A$ is a measurable function to the measurable space $\Delta A$ of all probability measures over $A$, but some readers may be unfamiliar with this notation.
- l. 114: $\Pi$ is not defined.
- l. 114: $F: \Pi \to \mathbb{R}$ is not limited to the (discounted) sum of rewards, right?
- l. 115: Is the discounting factor specified?
- l. 117--118: finding $R \in \mathcal{R}$, s.t. $\mathscr{A}\_M(R) = \arg\max\_{\pi \in \Pi}F(\pi)$?
- l. 120: $\mathcal{L}$ is not defined.
- l. 121: If we never need to train the language model, I think we can safely drop $\theta$ or use a more informative name.
- l. 124--125: How are $l_{feedback}$ and $l_{task}$ obtained?
- l. 125--126: $\max\_{l \in \mathcal{L}} F(\mathscr{A}\_M(p_\theta(l)))$ is weaker than $\max\_{R \in \mathcal{R}} F(\mathscr{A}\_M(R))$, which is weaker than finding $R \in \mathcal{R}$, s.t. $\mathscr{A}\_M(R) = \arg\max\_{\pi \in \Pi}F(\pi)$. There's a gap at each step. More rigorously, I think we should define the sequential decision-making problem formally.
- Eq. (1): I can guess $R \sim p\_\theta(x, z), z \sim p\_\theta(x)$ means using an LLM twice, but the notation is confusing.
- l. 205 and l.218: What does $\sim$ in $s_{node \sim \{elites\}}$ and $s_{node \sim \{tree\}}$ mean?
- Appendix I: maybe the author should show diff instead of just pasting all steps.

**Ethical Concerns:**

["NO or VERY MINOR ethics concerns only"]

**Final Justification:**

In the rebuttal, the author provided some empirical evidence regarding generalization ability.
If the results are valid, they demonstrate that the proposed method generalizes better than the baselines.
This design may be of interest to some readers in the RL community.

**Limitations:**

The author discussed the need for multiple interactions with the LLM and multiple RL training iterations.
However, I believe that the lack of evidence regarding generalization ability in novel or compositional tasks is a more significant limitation.

**Quality:**

3

**Strengths And Weaknesses:**

# Strengths

- Automated reward function design is an important problem.
- As far as I know, framing reward design as MCTS over LLM decisions is novel.
- The proposed method showed strong empirical results.
- The paper carefully evaluates contributions of different components (search, actions, self-verification).

# Weaknesses

## No test of generalization

All tasks are from predefined benchmarks with standard objectives (e.g., Ant: run forward). It's highly possible that there already exists manually designed reward functions in the training set of LLMs: Models such as GPT-4o and GPT-4o-mini have been trained on public code repositories and benchmarks, which very likely include environments like IsaacGym and Bi-DexHands, as well as papers, GitHub repos, and documentation that include reward functions for these tasks. There's no evidence that the proposed method works on novel or compositional tasks (e.g., Ant: run in a circle then reverse). Without such tests, we can't claim the method understands task descriptions in a generative or generalizable sense.

In my opinion, what RF-Agent demonstrates is search efficiency: it can find or tune reward functions more effectively than greedy (Eureka) or evolutionary (Revolve) baselines, but possibly only within the space the LLM already has internalized. It may be an efficient memory probe, not a general reward function generator.

## Limited understanding of feedback

The system feeds raw numerical logs $l_{feedback}$ into the LLM in string form. It assumes LLMs can reason over these logs to improve reward design, but this ability is unvalidated and possibly fragile. In my understanding, LLMs are not designed to do numerical trend analysis on raw sequences like a statistics model or a signal processor.

## Computational cost

Like other reward optimization methods, RF-Agent requires repeated LLM calls and full RL training loops. However, no analysis is provided on runtime, cost, or scalability, which is especially relevant given the use of GPT-4o.
I may have missed some information, but it seems that this paper only stated

> Overall, RF-Agent uses slightly more storage space and token consumption, but experimental results show that this slight sacrifice is worth it.

but gave no quantitative comparison.

---

> ### Author Rebuttal · Authors · 2025-07-27
>
> # Response to Reviewer qnw7
> We sincerely thank **Reviewer qnw7** for the detailed and comprehensive review. We particularly appreciate the acknowledgment that **framing reward design as MCTS over LLM decisions is novel** and for recognizing our **strong empirical results.** In response to the mentioned weaknesses and questions, we have organized our detailed replies into the following sections:
>
> ---
>
> ## Question 1: OOD tasks performance
> The reviewer's concern regarding generalization to novel tasks is indeed valid. To address this directly, we design and conduct two entirely new experiments using the Isaac Gym Ant environment:
>
> 1.  **'Ant Lie Down'**: This task tests for *generalization to a novel objective*. The instruction provided to the LLM is:
>     > "To make the ant lie down on its back (belly-up) as quickly and stably as possible. Its four feet should point towards the sky."
>
> 2.  **'Ant Patrol and Return'**: This is a *compositional task* designed to test the LLM's ability to generate reward functions for unforeseen, multi-stage problems. The instruction is:
>     > "To make the ant first run forward to the checkpoint position, then turn around and return to its starting origin position."
>
> The **evaluation score** for each task is based on a success condition, defined by an indicator function `1[·]`.
> * For **'Ant Lie Down'**, success is `1[orientation_reward > 0.9]`, corresponding to a stable belly-up pose.
> * For **'Ant Patrol and Return'**, success is `1[task_phase == 1 and dist_to_start < 0.5]`, signifying the completion of the entire two-stage trip.
>
> Using the same experimental configuration as in our main Isaac Gym experiments and employing the `GPT-4o-mini` model, we obtained the following results:
>
> | Task              | Sparse    | Eureka    | Revolve   | Ours      |
> |:------------------|:---------:|:---------:|:---------:|:---------:|
> | Lie Down          | 0.12±0.03 | 0.64±0.08 | 0.59±0.11 | 0.73±0.08 |
> | Patrol and Return | 0.05±0.02 | 0.52±0.07 | 0.63±0.05 | 0.82±0.06 |
>
> In these experiments, the 'Sparse' condition refers to using only the evaluation score as the reward signal. While it is not feasible to provide a handcrafted expert baseline for such novel tasks, the experimental results sufficiently demonstrate two key points:
>
> 1. The paradigm of LLM-based reward generation is capable of generalizing to these specialized, novel tasks.
> 2. Our RF-Agent continues to outperform the existing baseline methods on these new challenges.
>
> We will add this section to the manuscript and thank you again for this insightful question.
>
> ## Question 2: About Feedback Insight and Ablation
> We thank the reviewer for this insightful question, which highlights a critical and valid concern about the potential fragility of an LLM's numerical reasoning capabilities.
>
> We completely agree that LLMs are not inherently designed as numerical signal processors.
> As detailed in Appendix B.2 (line 567), this feedback is composed of three elements: nums, correspongding statistics, and analysis tip.
> We conducted the following ablation studies to verify the feedback effectiveness:
>
> * `w/o full feedback`: The baseline where `l_feedback` is entirely removed, and the search relies solely on the final success score, `F`.
> * `w/o nums`: Removes the numerical progression from the feedback.
> * `w/o statistics`: Removes the pre-computed statistical summary.
> * `w/o analysis tip`: Removes the analytical heuristic/tip.
>
> The results are as follows:
>
> | Task    | Ours (w/ full feedback) | w/o full feedback | w/o nums   | w/o statistics | w/o analysis tip |
> |:--------------|:-----------------------:|:-----------------:|:----------:|:--------------:|:----------------:|
> | ant   |   6.22   |  5.09(-18%)   | 5.28(-15%) |  5.63(-9%) |  5.47(-12%)  |
> | frankacabinet |  0.35  |  0.25(-28%)     | 0.32(-8%)  |   0.26(-25%)  |  0.31(-11%)  |
>
> Removing `l_feedback` entirely (`w/o full feedback`) causes a significant performance drop. More importantly, the ablations show that removing either the **pre-computed statistics** (`w/o statistics`) or the **analytical tip** (`w/o analysis tip`) individually degrades performance.
>
> We speculate that `analysis tip` helps LLM to a certain extent how to analyze sequences and further exert effects in subsequent diver actions. More refined optimizations can still be made regarding feedback, which can be discussed in depth in subsequent research, thank you again for your insights.
>
> ## Question 3: Computational Cost
> We tallied the `input_token` (prompt), `output_token` (completion), and the actual monetary cost for `RF-Agent`, `Eureka`, and `Revolve` on the Ant and Humanoid tasks. It is important to note that `input_tokens` can be processed in a single, efficient forward pass (a 'prefill'), **whereas `output_tokens` are generated autoregressively (a 'decode' process), which is more computationally intensive**. This efficiency difference is reflected in API pricing (e.g., Input: `$0.6 per 1M tokens` vs. Output: `$2.4 per 1M tokens`).
>
> | Ant          | Prefill Tokens | Decode Tokens | Total Cost ($) |
> |:----------------|---------------:|--------------:|---------------:|
> | Eureka   | 82458  | 54505  | 0.1802868|
> | Revolve   | 162348   | 43973   | 0.202944  |
> | **Ours-total** | **257564** | **69704** | **0.321828** |
> | &nbsp; ↳ Ours-action | 203282    | 53462   | -    |
> | &nbsp; ↳ Ours-others | 54282  | 16242  | -    |
>
>
> | Humaniod          | Prefill Tokens | Decode Tokens | Total Cost ($) |
> |:----------------|---------------:|--------------:|---------------:|
> | Eureka   | 78940    | 60021  | 0.191414   |
> | Revolve   | 169545   | 44118   | 0.20761 |
> | **Ours-total** | **254555** | **66557** | **0.31247** |
> | &nbsp; ↳ Ours-action | 204283   | 50270  | -     |
> | &nbsp; ↳ Ours-others | 50272  | 16287  | -    |
>
> The data in the tables leads to the following conclusions:
>
> 1.  `RF-Agent` utilizes **2-3 times more `prefill` tokens** than the baselines, while the number of `decode` tokens is only moderately higher (approx. **1.2-1.4 times**). Given that the `prefill` stage is computationally cheaper than the `decode` stage, the overall increase in resource consumption, as reflected by the total cost, is moderate and remains within an acceptable range.
>
> 2.  Furthermore, we find that this asymmetric token consumption aligns perfectly with the design philosophy of `RF-Agent`. The higher usage of `prefill` tokens is a direct result of leveraging historical information (i.e., previously generated thoughts and reward functions). As the MCTS tree grows, these artifacts are flexibly combined and included in the prompt, forming a **multi-step reasoning process** for the LLM. This process enables the LLM to generate more sophisticated and effective reward functions, which in turn leads to the **significant performance growth** observed in our experiments.
>
>
> ## Other Questions
> **Re l. 50-52:** On one hand, our ablation studies demonstrate the importance of balancing exploration and exploitation. On the other hand, a key reference is "The 2014 General Video Game Playing Competition," a benchmark for complex decision-making games. In this article, methods based on an exploration-exploitation balance, such as MCTS, consistently achieved top ranks (e.g., 1st, 2nd, 3rd), while approaches like genetic and greedy algorithms ranked significantly lower (e.g., 8th, 9th, 12th).
>
> **Re l. 112:** Yes. To maintain consistency with prior work, we assume the reward `R` is a function of the current state and action (`S x A`).
>
> **Re l. 113:** Thank you for the suggestion. We will change the notation for the action space to **A** for better readability.
>
> **Re l. 114 (π):** We will clarify in the text that `π` is a policy sampled from the policy space `Π`.
>
> **Re l. 114 (F):** Yes. `F` is a general evaluation score. It can represent the total sum of rewards or, for long-horizon tasks, a final binary signal (0/1) indicating task success.
>
> **Re l. 115:** Due to space constraints, we did not detail the full MDP definition in the main text. The discount factor `γ` is indeed part of the standard MDP formulation. We will add a detailed explanation to the appendix in the revised version.
>
> **Re l. 117-118:** Your understanding of the formulation is correct.
>
> **Re l. 120:** Thank you. We will add a definition clarifying that `ℒ` is the space of possible language sequences.
>
> **Re l. 121:** Thank you. We chose to retain the parameter `θ` to explicitly represent that different LLMs could be used in our framework, but we agree that it can be simplified for clarity.
>
> **Re l. 124-125:** `l_feedback` consists of a predefined text template combined with numerical values returned during the training process, as detailed in Appendix B.2 (line 567). `l_task` is determined by the benchmark itself, such as the brief text descriptions provided for each task in the Isaac Gym repository.
>
> **Re l. 125-126:** This is a very thoughtful point. In the revised manuscript, we will more formally define the high-level decision-making problem to ensure it strictly aligns with our proposed solution process.
>
> **Re Eq. (1):** This notation describes a single generation process where the LLM's response contains both the thought `z` and the reward function `R`. The model first decodes `z`, and then subsequently decodes `R`.
>
> **Re l. 205 and l. 218:** These terms refer to sampling nodes from the elite set and from the current MCTS tree, respectively. The information from these sampled nodes is then used for the next generation step.
>
> **Re Appendix I:** Thank you for this suggestion. For readability, we initially provided a one-sentence summary comparing the reward functions. We acknowledge that this may lack sufficient detail. In the revised manuscript, we will highlight the specific differences in the code using color.

---

> > ### Comment · Reviewer_qnw7 · 2025-08-04
> >
> > Thank you for your reply.
> >
> > If the new experimental results are valid, they can indeed support the generalization ability and practical benefits of the proposed design. Please incorporate them into the draft.
> >
> > I have no further questions.

---

> > > ### Author Response · Authors · 2025-08-05
> > >
> > > We thank reviewer qnw7 again for the constructive discussion. We will be sure to incorporate the new generalization experiments and address all other discussed points in the revised manuscript.

---

### Official Review · Reviewer_veGp · 2025-07-02

**Clarity:** 3
**Significance:** 3
**Originality:** 3
**Rating:** 5
**Confidence:** 3

**Summary:**

The paper proposes RF-Agent, a novel framework for automated reward function design in low-level control tasks. The core innovation lies in treating large language models (LLMs) as language agents and framing reward function design as a sequential decision-making process enhanced by Monte Carlo Tree Search (MCTS). This approach aims to address the limitations of existing methods, such as poor utilization of historical feedback and inefficient search, which hinder performance in complex control tasks. Experimental evaluations across 17 tasks demonstrate that RF-Agent outperforms state-of-the-art LLM-based methods, achieving higher success rates and training efficiency.

**Questions:**

### 1. Token Consumption Comparison

Given that RF-Agent's MCTS process requires the LLM to generate more reward functions than baselines like Eureka and Revolve, have the authors compared the number of tokens generated by the LLM for each method? Understanding the token consumption is essential for evaluating the practicality and cost-effectiveness of RF-Agent, especially when using expensive LLM models.

### 2. Comparison with Roska and Progress Function

Would the authors consider comparing RF-Agent with methods like Roska and Progress Function? Roska's continuous learning and reward evolution during policy improvement, as well as Progress Function's use of LLMs to model task progress, present alternative approaches to reward design. Such comparisons could shed light on the strengths and weaknesses of RF-Agent in different scenarios.

### 3. Component Ablation Details

The proposed method is comprised of many intricate designs for different purposes; however, they are not comprehensively analyzed. Could the authors elaborate on how each component, ranging from the score design to the action types, influences performance in different task domains?

### 4. Thought-Align Mechanism Effectiveness

The thought-align mechanism aims to resolve inconsistencies between LLM-generated design thoughts and reward function code. Could the authors provide quantitative results on the effectiveness of this mechanism? For example, what percentage of reward functions show improved alignment after thought-align, and how does this impact training performance?

**Ethical Concerns:**

["NO or VERY MINOR ethics concerns only"]

**Final Justification:**

I believe this work features solid experimental validation, and the manuscript would be more comprehensive if updated with this new content. While it incurs higher LLM interaction overhead and lacks direct numerical comparisons with some related methods (e.g., Roska), I recognize its rightful contributions within its specific research branch. Therefore, I have decided to increase the score.

**Limitations:**

yes.

**Quality:**

3

**Strengths And Weaknesses:**

## Strengths

### 1. Innovative Framework Combining LLMs and MCTS

The paper presents a compelling integration of LLMs and MCTS, treating reward function design as a sequential decision problem. This approach effectively leverages LLMs' contextual reasoning abilities and MCTS' search optimization to address the limitations of greedy or evolutionary algorithms used in previous work. The tree-based structure allows for systematic exploration and exploitation of reward function spaces, enhancing the utilization of historical information.

### 2. Comprehensive Experimental Validation

RF-Agent is evaluated across a diverse set of 17 low-level control tasks, covering locomotion and complex manipulation in IsaacGym and Bi-DexHands. The results show consistent superiority over baselines (Eureka, Revolve) in terms of success rates and training efficiency, even with lightweight LLM backbones like GPT-4o-mini. The quantitative evaluations, including average normalized scores and training curves, provide robust support for the method's effectiveness.

### 3. Clear Visualization and Comparison

The paper includes informative figures (e.g., Fig. 1, Fig. 2) that effectively illustrate the sequential decision-making view, method comparisons, and RF-Agent's tree search process. These visualizations help readers understand the differences between RF-Agent and baselines, as well as the step-by-step optimization path of reward functions. The comparison tables (e.g., Table 1) clearly present performance metrics, making it easy to assess the method's advantages.

### 4. Thoughtful Design of Action Types and Reasoning Mechanisms

RF-Agent introduces a variety of action types (mutation, crossover, path reasoning, different thought) to guide LLM-generated reward functions, each serving a specific purpose in exploring or exploiting the search space. The self-verify and thought-align mechanisms help mitigate LLM hallucinations and ensure alignment between design thoughts and reward function implementations, enhancing the framework's reliability.

## Weaknesses

### 1. Incomplete Ablation Studies

While the paper conducts ablation studies on the search method, action types, and reasoning components, the evaluation of some action types and their combinations is insufficient. For example, the specific impact of each action type (e.g., crossover vs. path reasoning) on different task types is not thoroughly analyzed. The ablation on "reasoning" only removes self-verify and thought alignment, but does not explore the individual contributions of each reasoning component.

### 2. Lack of Comparison with Relevant Methods

The paper does not compare RF-Agent with methods like Roska [1], which uses continuous learning and reward function evolution within a policy improvement process, or Progress Function [2], which employs LLMs to model task completion progress as an intrinsic bonus. These comparisons would help situate RF-Agent within the broader landscape of reward function design and highlight its unique contributions.

[1] Efficient Language-instructed Skill Acquisition via Reward-Policy Co-evolution

[2] Automated Rewards via LLM-Generated Progress Functions

### 3. Limited Analysis of Token Consumption and Compute Resources

Although the paper mentions that RF-Agent uses slightly more storage and tokens than baselines, it does not provide detailed quantitative comparisons of token consumption or compute resources. Given that MCTS requires generating more reward functions (and thus more LLM calls) than greedy or evolutionary methods, understanding the trade-off between performance and computational cost is crucial for practical adoption.

---

> ### Author Rebuttal · Authors · 2025-07-27
>
> # Response to Reviewer veGp
> We sincerely thank **Reviewer veGp** for the clear and detailed review. We particularly appreciate the positive feedback on our work, especially regarding the **innovative framework combining MCTS and LLMs** and the **clear figures and thorough experimental comparisons.** In response to the mentioned weaknesses and questions, we have organized our detailed replies into the following sections:
>
> ---
>
> ## Question 1: Token Consumption Comparison
> We tallied the `input_token` (prompt), `output_token` (completion), and the actual monetary cost for `RF-Agent`, `Eureka`, and `Revolve` on the Ant and Humanoid tasks. It is important to note that `input_tokens` can be processed in a single, efficient forward pass (a 'prefill'), **whereas `output_tokens` are generated autoregressively (a 'decode' process), which is more computationally intensive**. This efficiency difference is reflected in API pricing (e.g., Input: `$0.6 per 1M tokens` vs. Output: `$2.4 per 1M tokens`).
>
> |  Ant         | Prefill Tokens | Decode Tokens | Total Cost ($) |
> |:----------------|---------------:|--------------:|---------------:|
> | Eureka          | 82458          | 54505         | 0.1802868      |
> | Revolve         | 162348         | 43973         | 0.202944       |
> | **Ours-total** | **257564** | **69704** | **0.321828** |
> | &nbsp; ↳ Ours-action | 203282         | 53462         | -              |
> | &nbsp; ↳ Ours-others | 54282          | 16242         | -              |
>
>
> |  Humanoid         | Prefill Tokens | Decode Tokens | Total Cost ($) |
> |:----------------|---------------:|--------------:|---------------:|
> | Eureka          | 78940          | 60021         | 0.191414       |
> | Revolve         | 169545         | 44118         | 0.20761        |
> | **Ours-total** | **254555** | **66557** | **0.31247** |
> | &nbsp; ↳ Ours-action | 204283         | 50270         | -              |
> | &nbsp; ↳ Ours-others | 50272          | 16287         | -              |
>
> The data in the tables leads to the following conclusions:
>
> 1.  `RF-Agent` utilizes **2-3 times more `prefill` tokens** than the baselines, while the number of `decode` tokens is only moderately higher (approx. **1.2-1.4 times**). Given that the `prefill` stage is computationally cheaper than the `decode` stage, the overall increase in resource consumption, as reflected by the total cost, is moderate and remains within an acceptable range.
>
> 2.  Furthermore, we find that this asymmetric token consumption aligns perfectly with the design philosophy of `RF-Agent`. The higher usage of `prefill` tokens is a direct result of leveraging historical information (i.e., previously generated thoughts and reward functions). As the MCTS tree grows, these artifacts are flexibly combined and included in the prompt, forming a **multi-step reasoning process** for the LLM. This process enables the LLM to generate more sophisticated and effective reward functions, which in turn leads to the **significant performance growth** observed in our experiments.
>
> ## Question 2: Comparison with Roska and Progress Function(LLMCount)
> We thank the reviewer for suggesting these relevant works. We will cite and discuss the differences between these methods and our own in the revised manuscript.
>
> Due to the time constraints and the lack of publicly method code for reproduction, our comparison is presented in a **table** based on the differences and the results reported in their respective papers. For `ROSKA`, we noted that the number of training steps used for the Isaac Gym tasks differed from our setup. We retrained the reward function under the same steps for comparison.
>
> | Category   | Metric     | Eureka       | Revolve    | LLMCount[2]      | ROSKA[1]       | (Ours) RF-Agent         |
> |:----------------------|:------------------------|:-------------:|:---------------:|:----------------:|:--------------------------------|:--------------------------------------------------------------|
> | **knowledge requirement** | Environment Observation | Yes     | Yes      | Yes   | Yes         | Yes    |
> |                       | Additional Library   | No            | No     | Yes    | No     | No              |
> | **optimization method** | has optimization?       | Yes     | Yes     | No       | Yes       | Yes     |
> |                       | optimization method     | Greedy based  | Evolution based | -       | Greedy based      | exploration and exploitation tree search      |
> |                       | historical utilization  | Nearly Single | Population    | -       | implicitly with policy fusion | explicitly with diverse action by reasoning to llm |
> | **Performance** | normal locomotion     | Standard      | Medium  | No test in paper | Strong     | Stronger   |
> |            | normal manipulation    | Standard      | Medium     | No test in paper | Stronger    | Strong    |
> |          | complex manipulation   | Standard      | Medium  | Medium  | No test in paper   | Strong  |
>
> Based on the reviewer's valuable suggestions and our further analysis, we believe the following points are promising directions for future exploration:
>
> 1.  `LLMCount` focuses on achieving competitive results non-iteratively through reward transformation. This makes it highly efficient under very limited sample budgets, but it relies on the construction of external knowledge. For iterative optimization processes like ours, we believe the approach from `LLMCount` could serve as an **excellent initialization method**.
>
> 2.  Regarding the use of historical information, `ROSKA` **implicitly** fuses it at the **policy level** to assist its evolutionary process. In contrast, our method **explicitly** leverages history at the **LLM level**, using a rich set of actions and tree search to construct new reward functions. Each approach has its own advantages and is driven by different motivations. Our work, in particular, has been validated with high performance in more complex manipulation environments. **Future research could potentially combine these two distinct optimization paradigms.**
>
> ## Q3 & Q4: About Detailed Ablation
> We thank the reviewer for the valuable suggestion to conduct a more detailed ablation study. Given the rebuttal time constraints , we  include several key ablations as follows.
>
> ### Part1: More Ablation on Feedback
> Due to the rebuttal character limits, please check our reply to R2 (zhzV), located in part2 of Q2 for detailed ablation part of the feedback.
>
> ### Part2: More Ablation on Action Combination
>
> We further conducted a combinatorial study of action, which supplemented the ablation in the paper(line 324):
>
> | Action Combination                      | Ant         | Allegro Hand |
> |:----------------------------------------|:-----------:|:------------:|
> | a w/ `mutation`                           | 5.13(-18%)  | 18.76(-16%)  |
> | b w/ `mutation` + `crossover`             | 5.77(-7%)   | 21.90(-3%)   |
> | c w/ `mutation` + `reasoning`             | 5.94(-4%)   | 21.26(-5%)   |
> | d w/ `crossover` + `reasoning`            | 4.85(-22%)  | 19.21(-15%)  |
> | e w/ `mutation` + `crossover` + `reasoning` | 6.08(-2%)   | 22.19(-2%)   |
> | **Full Version (w/ all actions)** | **6.22** | **22.52** |
>
> By analyzing the ablation results, we can categorize our actions from the perspective of information utilization. The `mutation` action can be seen as a *local* operator, as it modifies a single parent node. In contrast, `crossover` and `reasoning` act as *global* operators, as they synthesize information from multiple nodes or entire ancestral paths.
>
> The results clearly indicate that relying exclusively on one type of operator leads to a significant performance drop. For instance, using only a *local* action (w/ `mutation`) or only *global* actions (w/ `crossover` + `reasoning`) both result in considerable performance degradation. Therefore, a balanced combination of both local and global operators is crucial for achieving the best performance.
>
> ### Part3: More Ablation on Thought-Align
>
> Finally, we present an ablation study on the `thought align` mechanism. In our current design, this process consists of two steps:
>
> 1.  **First Design Thought**: Following a Chain-of-Thought (CoT) like process, the LLM first generates a high-level 'design thought' *before* composing the actual reward function code.
> 2.  **Second Thought Align**: To mitigate potential hallucinations from a single generation step, after the reward function is generated, we prompt the LLM to re-examine the code's content and then revise the original 'design thought' to ensure it accurately reflects the code's logic.
>
> To quantitatively validate the effectiveness of this mechanism, we conducted further ablations on these intermediate components:
>
> | Task         | w/ full thought align | w/o first design thought | w/o second thought align | w/o all    |
> |:-------------|:---------------------:|:------------------------:|:------------------------:|:----------:|
> | ant          |          6.22         |       5.08(-18%)         |        5.94(-4%)         | 4.96(-20%) |
> | allegro hand |         22.52         |       18.07(-19%)        |       18.55(-17%)        | 14.64(-35%)|
>
> The results indicate that the `design thought` mechanism is crucial across all tasks. Notably, the **second `thought align` step has a particularly significant impact** on the more complex manipulation task (Allegro Hand).
>
> Regarding a direct percentage validation of this alignment's impact, it is challenging to isolate. This is because the second step corrects the thought, not the reward function code itself. The benefit of this corrected thought is realized later, when it is used to generate subsequent reward functions for child nodes. However, we are confident that this alignment process positively influences the quality of future rewards generated during the RF-Agent tree search, **as empirically demonstrated by the overall performance gains shown in the table above.**

---

> > ### Comment · Reviewer_veGp · 2025-08-05
> >
> > Thank you for your reply. Based on the content of your response, I believe this work features solid experimental validation, and the manuscript would be more comprehensive if updated with this new content. While it incurs higher LLM interaction overhead and lacks direct numerical comparisons with some related methods (e.g., Roska), I recognize its rightful contributions within its specific research branch. Therefore, I have decided to increase the score. At the same time, I hope that in future work, the authors will consider reducing interaction overhead and exploring comparisons and integration with other research branches.

---

> > > ### Author Response · Authors · 2025-08-05
> > >
> > > We sincerely thank the reviewer veGp for recognizing the value of our work. We will be sure to incorporate the content and new results from our rebuttal into the revised manuscript. Furthermore, we appreciate the insightful suggestions for future research, which we will certainly explore in our ongoing work.

---

### Official Review · Reviewer_zhzV · 2025-07-02

**Clarity:** 3
**Significance:** 2
**Originality:** 2
**Rating:** 5
**Confidence:** 3

**Summary:**

The paper presents RF-agent a method to generate (dense) reward functions using LLMs via sequential-decision making process based on  MCTS. While the contemporary approaches focus on LLM-based methods that rely on evolution-based methods or greedy-search based methods, this work proposes a tree based method that utilizes as MCTS and UCT to handle the exploration-exploitation process efficiently and generate better performing reward functions compared to the existing methods.

The LLM acts as a reward function generator that generates the reward function in python. There is another evaluation metric $F(\cdot)$ that captures how well a policy $\pi$ performs on a task, along with some feedback $l_{feedback}$ that indicates the feedback from executing the generated reward function on the task. The problem is to integrate this feedback in a sequential manner, so as to find better performing reward functions using the available actions (mutation, crossover, reasoning traces, etc.). This results in better ability to design dense reward functions from the sparse evaluation metric - compared to human designers or other baselines - in the low-level control simulation-based gym environments.

**Questions:**

As I mentioned in the weakness section, I'm mostly concerned about two things regarding the $l_{feedback}$: (1)  How well this method scales on OOD tasks and reward functions; and (2) any ablation or insights into it's design. I believe the work can be improved by addressing these concerns, and would convince me to change my score.

**Ethical Concerns:**

["NO or VERY MINOR ethics concerns only"]

**Final Justification:**

After discussion with the authors, my concerns related to data contamination and $l_{feedback}$ design and ablations were sufficiently addressed, so I'm updating my score to reflect that.

**Limitations:**

yes

**Quality:**

3

**Strengths And Weaknesses:**

# Strength
- The paper is well written and easy to follow.
-  Strong empirical results over the baselines that shows the promise of approaching this problem from the sequential decision-making lens.
- Rigorous experimental procedure and evaluation .

# Weakness
- Some of the motivation for the problem setting is not clear. For instance, why can't the evaluation function/metric and the feedback be directly used for optimizing the policy here? The goal is to find the best performing policy so it's not clear why we need an additional step to map $F \rightarrow R$, and then utilize the feedback for $R$ ? From the experiments, it seems the idea is to transform $F$ (spare reward) to $R$ (dense reward). It seems like both action space design and the feedback function design are the key component for the success of this method. The $l_\{feedback}$ is the crucial component that makes the search process possible - the space of possible LLM generated reward function is infinite and only this signal guides the algorithm in the right direction. What are the assumptions here? How to design such feedback function is an important aspect which is overlooked in the current draft.

- Another concern I have is that the methods/environments might already be in the training-set of the closed source LLMs like GPT, so there is a risk of data contamination. When you take this into account with the feedback prompt, maybe that can be a factor in the success of this method (as well as the baselines). How does this perform on OOD tasks or novel reward functions?

- There is ablation for $A$, that demonstrates these actions are justified for the domains in this work, but the paper needs a more thorough discussion on how to design this to make the work more generalizable.

---

> ### Author Rebuttal · Authors · 2025-07-27
>
> # Response to Reviewer zhzV
> We sincerely thank **Reviewer zhzV** for the insightful and detailed review. We particularly appreciate the acknowledgment that RF-Agent, through **rigorous experiments, provides promising support for solving this problem from a sequential decision-making perspective.** In response to the mentioned weaknesses and questions, we organize our detailed replies into the following sections:
> ___
>
> ## Question 1: OOD tasks performance
> The reviewer's concern regarding generalization to novel tasks is indeed valid. To address this directly, we design and conduct two entirely new experiments using the Isaac Gym Ant environment:
>
> 1.  **'Ant Lie Down'**: This task tests for *generalization to a novel objective*. The instruction provided to the LLM is:
>     > "To make the ant lie down on its back (belly-up) as quickly and stably as possible. Its four feet should point towards the sky."
>
> 2.  **'Ant Patrol and Return'**: This is a *compositional task* designed to test the LLM's ability to generate reward functions for unforeseen, multi-stage problems. The instruction is:
>     > "To make the ant first run forward to the checkpoint position, then turn around and return to its starting origin position."
>
> The **evaluation score** for each task is based on a success condition, defined by an indicator function `1[·]`.
> * For **'Ant Lie Down'**, success is `1[orientation_reward > 0.9]`, corresponding to a stable belly-up pose.
> * For **'Ant Patrol and Return'**, success is `1[task_phase == 1 and dist_to_start < 0.5]`, signifying the completion of the entire two-stage trip.
>
> Using the same experimental configuration as in our main Isaac Gym experiments and employing the `GPT-4o-mini` model, we obtained the following results:
>
> | Task              | Sparse    | Eureka    | Revolve   | Ours      |
> |:------------------|:---------:|:---------:|:---------:|:---------:|
> | Lie Down          | 0.12±0.03 | 0.64±0.08 | 0.59±0.11 | 0.73±0.08 |
> | Patrol and Return | 0.05±0.02 | 0.52±0.07 | 0.63±0.05 | 0.82±0.06 |
>
> In these experiments, the 'Sparse' condition refers to using only the evaluation score (i.e., the `1[...]` indicator function) as the reward signal. While it is not feasible to provide a handcrafted expert baseline for such novel tasks, the experimental results sufficiently demonstrate two key points:
>
> 1. The paradigm of LLM-based reward generation is capable of generalizing to these specialized, novel tasks.
> 2. Our RF-Agent continues to outperform the existing baseline methods on these new challenges.
>
> We will add this section to the manuscript and thank you again for this insightful question.
>
> ## Question 2: About Feedback Insight and Ablation
> ### Part1：Illustration on Feedback
> Regarding `l_feedback`, in current design, it serves as auxiliary information that summarizes the dynamics of the variables within a given reward function. As detailed in Appendix B.2 (line 567), this feedback is composed of three parts:
>
> 1. The change trend of each reward component over the training epochs.
> 2. Their corresponding statistics (e.g., max, mean, min).
> 3. Potential analytical suggestions.
>
> This design is intended to provide a moderate, not excessive, amount of information. In our framework, *the ideal optimization signal is the final task success rate, `F`*. The `l_feedback` is treated as a secondary, auxiliary factor to help analyze the reward's behavior. This choice is motivated by *practical deployment considerations, where a full state feedback might not be available*. Consequently, we deliberately limit the auxiliary information to only include the dynamics of the reward components themselves.
>
> To give a concrete example of how `l_feedback` influences the search process: *if a reward component exhibits very little variance during training, `l_feedback` suggests that this component might be ineffective*. This information then guides the node's action expansion. For instance, during a `crossover` operation between two high-performing reward functions (both with high `F`), the `l_feedback` allows the agent to identify and discard ineffective components from both parents. **By combining only the proven, meaningful components, the crossover is more likely to produce a superior child reward function.**
>
> Therefore, our current framework **places a greater emphasis on the design of the search `actions`** . We believe that well-designed `actions` are the primary mechanism for intelligently utilizing the insights from `l_feedback`.
>
> ### Part2: Ablation on Feedback
> Furthermore, the current design of `l_feedback` component has been experimentally validated to demonstrate its effectiveness. We conducted a set of ablation studies on its constituent parts:
>
> * `w/o full feedback`: A baseline condition where `l_feedback` is removed entirely. The MCTS optimization, as well as the information provided to the search actions, relies solely on the final evaluation score, `F`.
> * `w/o nums`: In this ablation, we remove the part of the feedback corresponding to the numerical progression of the reward components over the training epochs.
> * `w/o statistics`: This condition removes the corresponding statistical summary (e.g., max, mean, min) of the reward components.
> * `w/o analysis tip`: This condition removes the analytical "tip" which is intended to help the LLM interpret the feedback.
>
> We use the same ablation setting as in the original text and conducted the experiment, and the results are as follows:
>
> | Task          | Ours (w/ full feedback) | w/o full feedback | w/o nums   | w/o statistics | w/o analysis tip |
> |:--------------|:-----------------------:|:-----------------:|:----------:|:--------------:|:----------------:|
> | ant           |           6.22          |    5.09(-18%)     | 5.28(-15%) |   5.63(-9%)    |    5.47(-12%)    |
> | frankacabinet |           0.35          |    0.25(-28%)     | 0.32(-8%)  |   0.26(-25%)   |    0.31(-11%)    |
>
> The ablation study demonstrates that removing the `l_feedback` component entirely causes a significant performance degradation. However, it is noteworthy that even without this detailed feedback, the combination of `MCTS` and `LLM` is robust enough to achieve reasonable performance by optimizing solely on the final evaluation score.
>
> Furthermore, the results indicate that **each of the three auxiliary information components** within our `feedback` design contributes positively to the final performance.
>
> We will include a more in-depth discussion of these findings in our revised manuscript.
>
> ## Question 3: About Reward Design Problem Motivation
>
> We thank the reviewer for this insightful question, as it allows us to further elaborate on the motivation behind our **reward design problem setting**. This approach originates from the principles of reward engineering. The evaluation score, `F`, often serves as a sparse and monolithic signal. Using `F` directly as the reward `R` for policy learning is often inefficient and makes the learning process difficult, especially for complex manipulation and compositional tasks.
>
> This challenge is particularly evident in our experimental results.
> * As shown by the `Sparse` results for `Bi-DexHands` in *Figure 3(after line 296)*, when a task only provides a binary 0/1 success signal at the end of an episode, a policy trained directly on this sparse reward can barely learn within the given step limit.
> * Similarly, as shown in *Table 1(after line 248)*, even for tasks like `Ant` and `Humanoid` that provide a dense metric `F` at every step, using `F` directly as the reward is still significantly less sample-efficient compared to using a shaped reward function.
>
> Therefore, the significance of our problem setting is to explore **how to better construct dense reward signals to efficiently guide policy learning**. From this perspective, a reward function is an interpretable, easily deployable, and modifiable way to provide this guidance. Furthermore, our method investigates how to leverage `LLMs` to generate these better reward functions. This approach enables automated optimization while **reducing the laborious manual effort** required from human experts in reward engineering.
>
> ## Question 4: About Action Design Discussion
> The design of our current `action` framework is intentionally high-level and conceptual. It does not operate on concrete environmental assets or objects (e.g., it does not propose adding a specific 'velocity penalty'). Instead, our action set can be applied to any reward function that exhibits fundamental characteristics, such as using a selection of environment observations, containing tunable hyperparameters, and being composed of multiple components. **This makes our framework inherently generalizable.**
>
> In designing this framework, we integrated several key ideas: common heuristic operations (like `mutation` and `crossover`), an action that simulates human-like step-by-step thinking (`reasoning`), and an action that emphasizes exploration (`different thought`). Together, these form a comprehensive set of high-level, conceptual actions designed to effectively navigate the complex space of reward functions.

---

> > ### Comment · Reviewer_zhzV · 2025-08-06
> > **Response**
> >
> > Thanks for the answering my questions and sharing more details. Some of my questions regarding OOD and feedback design have been answered. However, the arguments for Question 3 and 4 can still benefit from more evidence (across different environments/domains).
> >
> > Overall, I think these details will improve the manuscript so I'm revising my score to reflect that.

---

> > > ### Author Response · Authors · 2025-08-06
> > >
> > > We sincerely thank the reviewer zhzV for the positive feedback and valuable discussion. We appreciate the suggestions for further improvement and will be sure to incorporate more evidence in the revised manuscript.

---

### Official Review · Reviewer_87TL · 2025-07-03

**Clarity:** 3
**Significance:** 3
**Originality:** 3
**Rating:** 4
**Confidence:** 4

**Summary:**

This paper explores a novel method of using Large Language Models (LLMs) to generate reward functions in reinforcement learning (RL). Considering that current methods underutilize historical feedback and the optimization capabilities of LLMs, this paper proposes a systematic framework that uses LLM to generate and update reward functions with the help of Monte Carlo Tree Search (MCTS) . The paper employs MCTS to make decisions and update the reward functions, and emphasizes the role of action expansion in reshaping the reward functions. Experimental results on low-level control tasks demonstrate that the proposed approach can generate high-performance reward functions, thereby facilitating policy learning.

**Questions:**

-Building on the above weaknesses 2, while the motivation for combining MCTS with reward optimization is reasonable, it is unclear how many iterations of reward function updates are required for the tasks in the experiments to achieve stable success. Overall, for training a task from start to successful execution, what is the RL training cost? How does the total training time compare to other methods? Is it within a reasonable and acceptable range?
-In Figure 2, does the process from s_1^0 to its leaf nodes s_2^0 and s_2^1 represent action expansion? If not, how does it differ from action expansion?
-Is there a possibility to provide a detailed explanation of the selection phase? When a node is selected, will other nodes at the same level also be visited later during the search? My understanding is that during the backpropagation phase, all nodes on this path are updated, after which other branches are explored until the reward function corresponding to a certain node successfully completes the task. Is there a limit to the action expansion depth along a single path? Has the author experimented with different depth limits?
-Are there ablation studies on whether to use design thought? Regarding Weakness 1 mentioned above, is there a test that compares the impact of using different design thoughts on reward generation and optimization? Alternatively,  are there experiments with using Chain of Thought (CoT) reasoning to guide LLMs in generating reward functions?

**Ethical Concerns:**

["NO or VERY MINOR ethics concerns only"]

**Final Justification:**

The authors took the feedback seriously and improved their work accordingly. The remaining issues are relatively minor and shouldn't prevent acceptance, though addressing them would further strengthen the paper.

**Limitations:**

Yes

**Quality:**

3

**Strengths And Weaknesses:**

Strengths:

-The paper is well-written, clear, and easy to understand.
-The motivation for introducing MCTS to optimize reward functions is natural, so that the training effects corresponding to various reward functions can be quickly listed and backtracked, with optimization at the nodes. The paper demonstrates that under this design, it is possible to arouse the potential of LLM for efficiently designing reward functions.
-The design of action expansion stimulates the LLM to improve existing reward functions from multiple perspectives, making better use of historical information and potential directions for optimizing reward functions.
-Experimental results on low-level control tasks provide adequate evidence for the effectiveness of the proposed reward optimization framework and demonstrate its generalizability.

Weaknesses:

-LLM's design thoughts for reward generation lack a finer-grained structure. For example, when generating reward functions, LLMs could be guided to decompose the design thought into more granular components at the reward function level. This might include explicitly defining the set of usable variables, identifying which subsets contribute to specific components of the reward, and describing how these components combine into the overall reward. In other words, standardizing the design thought process could be an effective way to enhance LLM's search strategies. By doing so, during action expansion, the LLM could quickly backtrack to identify which variables and reward components were used or effective, enabling faster iteration and improving the explainability of the reward optimization process. A possible implementation is discussed in [1].
-Lack of analysis on MCTS iterations and computational cost. Specifically, the paper does not analyze the number of iterations or the time required for MCTS. It remains unclear how many rounds of RL training and reward updates are needed for a given task to achieve stable and successful execution. This raises concerns about the overall training cost and whether it falls within an acceptable range compared to alternative methods.
[1] Heng Z K, Zhao Z, Wu T, et al. Boosting Universal LLM Reward Design through Heuristic Reward Observation Space Evolution[J]. arXiv preprint arXiv:2504.07596, 2025.

---

> ### Author Rebuttal · Authors · 2025-07-27
>
> # Response to Reviewer 87TL
> We sincerely thank **Reviewer 87TL** for the thorough and constructive review. We particularly appreciate the recognition that **the motivation for introducing MCTS is natural and novel** and that **the experimental section sufficiently demonstrates the effectiveness** of our approach. In response to your mentioned weaknesses and questions, we consolidate our detailed replies into the following sections:
> ___
>
> ## Question 1: About MCTS Iterations
> We thank the reviewer for acknowledging the novelty of our MCTS-based approach to reward function optimization.
>
> To ensure a fair comparison, as stated on *line 278*, all methods were allocated an **identical number of total samples** per experiment. Each **sample** constitutes a full cycle of LLM-based reward generation and a subsequent RL training run. Consequently, since the number of samples is identical for all methods in each experiment, the upper bound on total training time is also consistent across all approaches.
>
> To further analyze efficiency, we can compare the training cost (i.e., number of samples) required to achieve stable performance on the more challenging Bi-DexHands tasks. As shown in *Figure 5 (after line 318)*, which plots success rate versus the number of samples, we can establish performance thresholds. Defining "stable success" at 0.6 and 0.8 success rates, the number of samples required by each method to reach these levels is as follows:
>
> | Metric  | Ours (Easy) | Eureka (Easy) | Revolve (Easy) | | Ours (Hard) | Eureka (Hard) | Revolve (Hard) |
> |:--------|:-----------:|:-------------:|:--------------:|-|:-----------:|:-------------:|:--------------:|
> | avg 0.6 |     108     |     ~512      |     ~512       | |     100     |      304      |      144       |
> | avg 0.8 |     264     |     512+      |     512+       | |     296     |     512+      |     512+       |
>
> In this table, `Easy/Hard` denote the `BiDex-Expert-Easy/Hard groups`. `~512` means a baseline was approaching the target success rate by the 512-sample limit, whereas `512+` means it failed to do so.
>
> The results clearly show that our method is more sample-efficient. On the Easy tasks, for example, it reaches a 0.6 success rate nearly five times faster than the baselines, which fail to reach the 0.8 success rate within the entire budget. Regarding the precise wall-clock duration, we refer the reviewer to *Appendix C (line 620)*. While the full training budget for each Bi-DexHands task is approximately 40 hours, our method consistently converges to a high success rate in under 20 hours.
>
> We hope these details resolve any concerns regarding training efficiency and time, and we conclude that RF-Agent is a more efficient method that achieves superior performance in less total training time.
>
> ## Question 2: About Aciton Expansion
> Your understanding is comprehensive. The transition from `s_1^0` to `s_2^0` and `s_2^1` is also an action expansion. The reason it is not marked with an "orange" arrow in *Figure 2(after line 126)* is to differentiate the stages of the search process.
>
> The non-orange arrows indicate expansions that were previously completed and whose resulting nodes (like `s_2^0` and `s_2^1`) are already stored in the tree. The orange arrow, however, is used to specifically denote the expansion (to node `s_5^1`) that is currently active. In this way, the figure illustrates a dynamic process where arrow colors distinguish past expansions from the present one.
>
> ## Question 3: About Selection Stage and Depth
> Your description about the selection phase is basically accurate. We would like to add an implementation detail for clarification. For ease of deployment, we perform action expansion in parallel from the selected leaf node.
>
> For instance, when the selection phase reaches a leaf node `s_1^0` (i.e., a node with no children), we simultaneously expand n new child nodes (`s_2^0`, `s_2^1`, ...), each corresponding to a different action. Once this batch of new nodes completes their simulations, their values are backpropagated up the tree (e.g., `s_2^0 -> s_1^0 -> root` and `s_2^1 -> s_1^0 -> root`), just as the reviewer described. The backpropagations are processed sequentially in the order that the simulations finish.
>
> In the subsequent selection phase, the search might proceed to a different branch (e.g., to sibling node `s_1^1`) if its UCT score is higher. Alternatively, if the backpropagated values have made the `s_1^0` branch more promising, the search may continue deeper into one of its newly expanded children (`s_2^0`, `s_2^1`, ...). This creates an efficient and flexible iterative optimization process.
>
> Regarding tree depth, our maximum depth setting is correlated with the total simulation budget (e.g., a depth limit of 16 for 512 total nodes). Empirically, we observed that the search tree's effective depth naturally stabilized between 8 and 12, which was sufficient to achieve high success rates. Constraining the depth to be **shallower** (<8) confirmed its importance, as it caused a significant **performance degradation** on some tasks (e.g., the success rate on Bidex-CatchAbreast dropped from 0.7 to 0.5).
>
> ## Question 4: About Design Thought
> We are grateful to the reviewer for this crucial point. In our experiments, the "design thought" serves as a high-level plan, generated by the LLM itself, for composing the reward function. We require this thought to explain how different components are combined into a whole, which is an approach **analogous to Chain-of-Thought** (CoT).
>
> We agree that the "design thought" process could be made more effective with a more structured framework. As the reviewer insightfully suggests with reference [1], utilizing a state memory to track which state variables are used for the reward function would indeed be a more effective approach.
>
> Inspired by this, we conducted a brief preliminary experiment with three conditions: (1) without "design thought" (**w/o thought**), (2) with our original self-generated "design thought" (**w/ self thought**), and (3) with a "state-structured thought" (**w/ state-structured thought**). The "state-structured thought" condition implements the Reward State Execution mechanism mentioned in [1]. In this setup, we track the historical usage of variables and prompt the LLM to generate a "thought" based on this auxiliary information, ensuring the thought specifies how the selected variables are to be combined.
>
> | Task          | w/o thought | w/ self thought | w/ state-structured thought |
> |:--------------|:-----------:|:--------------:|:--------------------------:|
> | ant           | 5.08(-18%)  |      6.22      |        6.43(+3%)         |
> | frankacabinet | 0.27(-23%)  |      0.35      |        0.46(+31%)        |
>
> The experimental results show that the "design thought" process is necessary, as its absence degrades reward design performance. Furthermore, a more fine-grained thought process, as implemented in the "state-structured thought" condition, yields additional performance improvements. The Reward State Execution mechanism is indeed analogous to an external memory module in LLM agent design. We are grateful for the reviewer's profound insight and will add a discussion of the performance gains from this structured approach to our revised manuscript.
>
> Regarding CoT reasoning, our original "self thought" method is, in fact, an implementation of this process. As formalized in *Equation 1(after line 165)*, our method first generates a "self thought"—the high-level reasoning plan—before composing the corresponding reward function. Therefore, the performance improvement observed when comparing the w/ self thought and w/o thought conditions already demonstrates the effectiveness of CoT reasoning in our framework.

---

> > ### Comment · Reviewer_87TL · 2025-08-07
> >
> > Thank the authors for this detailed rebuttal. It shows that the authors took the feedback seriously and improved their work accordingly. The remaining issues are relatively minor and shouldn't prevent acceptance, though addressing them would further strengthen the paper. For example, expand the structured thought experiments to more tasks, add statistical analysis to their claims, and discuss LLM API costs if applicable.

---

> > > ### Author Response · Authors · 2025-08-07
> > >
> > > We sincerely thank the reviewer 87TL for the positive feedback and constructive discussion. We will supplement the content of the rebuttal section in the manuscript. At the same time, regarding the analysis of the LLM API you mentioned, we have also analyzed it in the discussion of reviewer veGp and qnw7, and we will also add it to the manuscript. Finally, we thank reviewer 87TL again for the positive discussion.

---

### Note · Authors · 2025-08-15

We wish to express our sincere gratitude to the Area Chair and the reviewers for a highly constructive and thorough review process. The opportunity to clarify our work and respond to the excellent feedback has been immensely helpful, and we are pleased that the resulting discussion has resolved the main questions and led to positive reassessments of our work.

To summarize, our work, **RF-Agent**, makes three primary contributions:

1.  **A Novel Formulation for Reward Design:** Our work introduces a novel paradigm that frames the challenge of automatic reward generation as a sequential decision-making problem, solved by applying Monte Carlo Tree Search (MCTS) over the space of LLM-generated reward candidates.

2.  **A Novel and Comprehensive Search Framework:** RF-Agent features a sophisticated design, utilizing a rich set of conceptual actions, a reflective thought-align mechanism, and the full use of multi-stage contextual information inherent in the MCTS process.

3.  **Extensive Empirical Validation:** Our work is supported by extensive experiments across a wide range of challenging robotics environments, including both locomotion and complex manipulation. We provide a thorough comparison against state-of-the-art baselines and include detailed ablation studies that validate our key design choices.

Furthermore, in direct response to the reviewers' excellent suggestions during the rebuttal period, we conducted additional experiments on novel and compositional tasks to further demonstrate the generalizability, robustness, and overall effectiveness of our framework.

We believe that the dialogue during this review, combined with our additional clarifications and results, has solidified the standing of RF-Agent as a significant and practical contribution. Thank you once again for your time and expertise in helping us improve this manuscript.

---

### Decision · Program_Chairs · 2025-09-17

**Decision:**

Accept (spotlight)

**Comment:**

This works aims to find a reward function that can help RL agents optimize for a desired (sparse) goal. To do this, authors make use of LLMs to perform tree search in the reward space. Specifically, output of each node is a python program that specifies a reward function. The feedback obtained for each node corresponds to performance of training an agent with the respective reward function (and some additional stats). This leverages the long-memory and optimization capabilities of LLMs to sequentially update the reward function.

All the reviewers appreciated the novelty of the approach and the exhaustive empirical analysis. Some questions were raised pertaining to the cost and compute time, and also regarding ensuring a fair evaluation since the RL environments might have been already there in the LLM training data. Authors did a commendable job of providing all the details and running additional OOD experiments to affirm the utility of their approach.

Overall, the proposed method is novel, looks at an important problem, and the empirical analysis is thorough.